# A Survey of Temporal Credit Assignment in Deep Reinforcement Learning

**Eduardo Pignatelli**                                   *e.pignatelli@ucl.ac.uk*
*University College London*

**Johan Ferret**                                             *jferret@google.com*
*Google DeepMind*

**Hado van Hasselt**                                        *hado@google.com*
*Google DeepMind*

**Matthieu Geist**                                          *mfgeist@google.com*
*Google DeepMind*

**Thomas Mesnard**                                         *mesnard@google.com*
*Google DeepMind*

**Olivier Pietquin**                                        *pietquin@google.com*
*Google DeepMind*

**Laura Toni**                                                 *l.toni@ucl.ac.uk*
*University College London*

**Reviewed on OpenReview:** *https://openreview.net/forum?id=bNtr6SLgZf*

## Abstract

The Credit Assignment Problem (CAP) refers to the longstanding challenge of Reinforcement Learning (RL) agents to associate actions with their long-term consequences. Solving the CAP is a crucial step towards the successful deployment of RL in the real world since most decision problems provide feedback that is noisy, delayed, and with little or no information about the causes. These conditions make it hard to distinguish serendipitous outcomes from those caused by informed decision-making. However, the mathematical nature of credit and the CAP remains poorly understood and defined. In this survey, we review the state of the art of Temporal Credit Assignment (CA) in deep RL. We propose a unifying formalism for credit that enables equitable comparisons of state of the art algorithms and improves our understanding of the trade-offs between the various methods. We cast the CAP as the problem of learning the influence of an action over an outcome from a finite amount of experience. We discuss the challenges posed by *delayed effects*, *transpositions*, and a *lack of action influence*, and analyse how existing methods aim to address them. Finally, we survey the protocols to evaluate a credit assignment method and suggest ways to diagnose the sources of struggle for different methods. Overall, this survey provides an overview of the field for new-entry practitioners and researchers, it offers a coherent perspective for scholars looking to expedite the starting stages of a new study on the CAP, and it suggests potential directions for future research.

# 1   Introduction

RL is poised to impact many real world problems that require sequential decision making, such as strategy (Silver et al., 2016; 2018; Schrittwieser et al., 2020; Anthony et al., 2020; Vinyals et al., 2019; Perolat et al., 2022) and arcade video games (Mnih et al., 2013; 2015; Badia et al., 2020; Wurman et al., 2022), climate control (Wang & Hong, 2020), energy management (Gao, 2014), car driving (Filos et al., 2020) and stratospheric balloon navigation (Bellemare et al., 2020), designing circuits (Mirhoseini et al., 2020), cybersecurity (Nguyen & Reddi, 2021), robotics (Kormushev et al., 2013), or physics (Degrave et al., 2022). One fundamental mechanism allowing RL agents to succeed in these scenarios is their ability to evaluate the *influence* of their actions over outcomes – e.g., a win, a loss, a particular event, a payoff. Often, these outcomes are consequences of isolated decisions taken in a very remote past: actions can have long-term effects. The problem of learning to associate actions with distant, future outcomes is known as the temporal Credit Assignment Problem (CAP): *to distribute the credit of success among the multitude of decisions involved* (Minsky, 1961). Overall, the *influence* that an action has on an outcome represents *knowledge* in the form of *associations* between actions and outcomes (Sutton et al., 2011; Zhang et al., 2020). These associations constitute the scaffolding that agencies can use to deduce, reason, improve and act to address decision-making problems and ultimately improve their data efficiency.

Solving the CAP is paramount since most decision problems have two important characteristics: they take a *long time to complete*, and they seldom provide immediate feedback, but often *with delay* and little insight as to which actions caused it. These conditions produce environments where the feedback signal is weak, noisy, or deceiving, and the ability to separate serendipitous outcomes from those caused by informed decision-making becomes a hard challenge. Furthermore, as these environments grow in complexity with the aim to scale to real-world tasks (Rahmandad et al., 2009; Luoma et al., 2017), the actions taken by an agent affect an increasingly vanishing part of the outcome. In these conditions, it becomes challenging to learn value functions that accurately represent the *influence* of an action and to be able to distinguish and order the relative long-term values of different actions. In fact, canonical Deep Reinforcement Learning (Deep RL) solutions to *control* are often brittle to the hyperparameter choice (Henderson et al., 2018), inelastic to generalise zero-shot to different tasks (Kirk et al., 2023), prone to overfitting (Behzadan & Hsu, 2019; Wang et al., 2022), and sample-inefficient (Ye et al., 2021; Kapturowski et al., 2023). Overall, building a solid foundation of knowledge that can unlock solutions to complex problems beyond those already solved calls for better CA techniques (Mesnard et al., 2021).

In the current state of RL, *action values* are a key proxy for *action influence*. Values *actualise* a return by synthesising statistics of the *future* into properties of the *present*: they transform a signal dependent on the future into one dependent only on the present. Recently, the advent of Deep RL (Arulkumaran et al., 2017) granted access to new avenues to express credit through values, either by using memory (Goyal et al., 2019; Hung et al., 2019), associative memory (Hung et al., 2019; Ferret et al., 2021a; Raposo et al., 2021), counterfactuals (Mesnard et al., 2021), planning (Edwards et al., 2018; Goyal et al., 2019; van Hasselt et al., 2021) or by meta-learning (Xu et al., 2018; Houthooft et al., 2018; Oh et al., 2020; Xu et al., 2020; Zahavy et al., 2020). The research on CAP is now fervent, and with a rapidly growing corpus of works.

**Motivation.**   Despite its central role, there is little discussion on the precise mathematical nature of credit. While these proxies are sufficient to unlock solutions to complex tasks, it remains unclear where to draw the line between a generic measure of action influence and *credit*. Existing works focus on partial aspects or sub-problems (Hung et al., 2019; Arjona-Medina et al., 2019; Arumugam et al., 2021) and not all works refer to the CAP explicitly in their text (Andrychowicz et al., 2017; Nota et al., 2021; Goyal et al., 2019), despite their findings providing relevant contributions to address the problem. The resulting literature is fragmented and lacks a space to connect recent works and put their efforts in perspective for the future. The field still holds open questions:

**Q1.** *What* is the *credit* of an action? How is it different from an *action value*? And what is the CAP? What in words, and what in mathematics?

**Q2.** *How* do agents learn to *assign* credit? What are the main methods in the literature and how can they be organised?

**Q3.** How can we *evaluate* whether a method is improving on a challenge? How can we monitor advancements?

**Goals.** Here, we propose potential answers to these questions and set out to realign the fundamental issue raised by Minsky (1961) to the Deep RL framework. Our main goal is to provide an overview of the field to new-entry practitioners and researchers, and, for scholars looking to develop the field further, to put the heterogeneous set of works into a comprehensive, coherent perspective. Lastly, we aim to reconnect works whose findings are relevant for CAP, but that do not refer to it directly. To the best of our knowledge, the work by Ferret (2022, Chapter 4) is the only effort in this direction, and the literature offers no explicit surveys on the temporal CA problem in Deep RL.

**Scope.** The survey focuses on temporal CA in single-agent Deep RL, and the problems of *(i)* quantifying the influence of an action mathematically and formalising a mathematical objective for the CA problem *(ii)* defining its challenges, and categorising the existing methods to learn the quantities above, *(iii)* defining a suitable evaluation protocol to monitor the advancement of the field. We do not discuss *structural* CA in Deep Neural Networks (DNNs), that is, the problem of assigning credit or blame to individual parameters of a DNN (Schmidhuber, 2015; Balduzzi et al., 2015). We also do not discuss CA in multi-agent RL, that is, to ascertain which agents are responsible for creating good reinforcement signals (Chang et al., 2003; Foerster et al., 2018). When credit (assignment) is used without any preceding adjective, we always refer to *temporal* credit (assignment). In particular, with the adjective *temporal* we refer to the fact that *"each ultimate success is associated with a vast number of internal decisions"* (Minsky, 1961) and that these decisions, together with states and rewards, are arranged to form a *temporal* sequence.

The survey focuses on Deep RL. In surveying existing formalisms and methods, we only look at the Deep RL literature, and when proposing new ones, we tailor them to Deep RL theories and applications. We exclude from the review methods specifically designed to solve decision problems with linear or tabular RL, as they do not bode well for scaling to complex problems.

**Outline.** We address **Q1.**, **Q2.** and **Q3.** in the three major sections of the manuscript. Respectively:

- **Section 4** addresses **Q1.**, proposing a definition of credit and the CAP and providing a survey of action influence measures.

- **Section 5** and **Section 6** address **Q2.**, discussing the key challenges to solving the CAP and the existing methods to assign credit, respectively.

- **Section 7** answers **Q3.**, reviewing the problem setup, the metrics, and the evaluation protocols to monitor advancements in the field.

- **Section 8** summarises the main points of the manuscript and provides a critical discussion to highlight the open challenges.

For each question, we contribute by: *(a)* systematising *existing works* into a simpler, coherent space; *(b)* discussing it, and *(c)* synthesising our perspective into a unifying formalism. Table 1 outlines the suggested reading flow according to the type of reader.

| Reader type | Suggested Flow |
|---|---|
| Specialised CA scholar | $1 \to 8 \to 4 \to 5 \to 6 \to 7 \to 2$ |
| RL researcher | $1 \to 8 \to 4 \to 5 \to 6 \to 7$ |
| Deep Learning researcher | $1 \to 3 \to 4 \to 5 \to 6 \to 7 \to 8$ |
| Practitioner (applied researcher) | $6 \to 4.4 \to 3$ |
| Proposing a new CA method | $8 \to 7 \to 6 \to 2 \to 4$ |

Table 1: Suggested flow of reading by type of reader to support the outline in Section 1. Numbers represent section numbers.

## 2 Related Work

Three existing works stand out for proposing a better understanding of the CAP explicitly. Ferret (2022, Chapter 4) designs a conceptual framework to unify and study credit assignment methods. The chapter proposes a general formalism for a range of credit assignment functions and discusses their characteristics and general desiderata. Unlike Ferret (2022, Chapter 4), we survey potential formalisms for a mathematical definition of credit (Section 4); given the new formalism, we propose an alternative view of the methods to assign credit (Section 6), and an evaluation protocol to measure future advancements in the field. Arumugam et al. (2021) analyses the CAP from an information theoretic perspective. The work focuses on the notion of *information sparsity* to clarify the role of credit in solving sparse reward problems in RL. Despite the work questioning what credit is mathematically, it does not survey existing material, and it does not provide a framework that can unify existing approaches to represent credit under a single formalism. Harutyunyan et al. (2019) propose a principled method to measure the credit of an action. However, the study does not aim to survey existing methods to *measure* credit, the methods to *assign* credit, and the methods to evaluate a credit assignment method, and does not aim to organise them into a cohesive synthesis.

The literature also offers surveys on related topics. We discuss them in Appendix A to preserve the fluidity of the manuscript.

As a result, none of these works position CAP in a single space that enables thorough discussion, assessment and critique. Instead, we propose a formalism that unifies the existing *quantities* that represent the influence of an action (Section 4). Based on this, we can analyse the advantages and limitations of existing measures of action influence. The resulting framework provides a way to gather the variety of existing *methods* that learn these quantities from experience (Section 6), and to monitor the advancements in solving the CAP.

## 3 Notation and Background

Here we introduce the notation that we will follow in the rest of the paper and the required background.

**Notations.** We use calligraphic characters to denote sets and the corresponding lowercases to denote their elements, for example, $x \in \mathcal{X}$. For a measurable space $(\mathcal{X}, \Sigma)$, we denote the set of probability measures over $\mathcal{X}$ with $\Delta(\mathcal{X})$. We use an uppercase letter $X$ to indicate a random variable, and the notation $\mathbb{P}_X$ to denote its distribution over the sample set $\mathcal{X}$, for example, $\mathbb{P}_X : \mathcal{X} \to \Delta(\mathcal{X})$. When we mention a *random event X* (for example, a *random action*) we refer to a random draw of a specific value $x \in \mathcal{X}$ from its distribution $\mathbb{P}_X$ and we write, $X \sim \mathbb{P}_X$. When a distribution is clear from the context, we omit it from the subscript and write $\mathbb{P}(X)$ instead of $\mathbb{P}_X(X)$. We use $\mathbb{1}_{\mathcal{Y}}(x)$ for the indicator function that maps an element $x \in \mathcal{X}$ to 1 if $x \in \mathcal{Y} \subset \mathcal{X}$ and 0 otherwise. We use $\mathbb{R}$ to denote the set of real numbers and $\mathbb{B} = \{0, 1\}$ to denote the Boolean domain. We use $\ell_\infty(x) = \|x\|_\infty = \sup_i |x_i|$ to denote the $\ell$-infinity norm of a vector $x$ of components $x_i$. We write the Kullback-Leibler divergence between two discrete probability distributions $\mathbb{P}_P(X)$ and $\mathbb{P}_Q(X)$ with sample space $\mathcal{X}$ as: $D_{KL}(\mathbb{P}_P(X)||\mathbb{P}_Q(X)) = \sum_{x \in \mathcal{X}}[\mathbb{P}_P(x) \log(\mathbb{P}_P(x)/\mathbb{P}_Q(x))]$.

**Reinforcement Learning.** We consider the problem of learning by interacting with an environment. A program (the *agent*) interacts with an *environment* by making decisions (*actions*). The action is the agent's interface with the environment. Before each action, the agent may *observe* part of the environment and take suitable actions. The action changes the state of the environment. After each action, the agent may perceive a feedback signal (the *reward*). The goal of the agent is to learn a rule of behaviour (the *policy*) that maximises the expected sum of rewards.

**Markov Decision Processes (MDPs).** MDPs formalise decision-making problems. This survey focuses on the most common MDP settings for Deep RL. Formally, a discounted MDP (Howard, 1960; Puterman, 2014) is defined by a tuple $\mathcal{M} = (\mathcal{S}, \mathcal{A}, R, \mu, \gamma)$. $\mathcal{S}$ is a finite set of states (the *state space*) and $\mathcal{A}$ is a finite set of actions (*the action space*). $R : \mathcal{S} \times \mathcal{A} \to [r_{min}, r_{max}]$ is a deterministic, bounded reward function that maps a state-action pair to a scalar reward $r$. $\gamma \in [0, 1]$ is a discount factor and $\mu : \mathcal{S} \times \mathcal{A} \to \Delta(\mathcal{S})$ is a transition kernel, which maps a state-action pair to probabilities over states. We refer to an arbitrary state $s \in \mathcal{S}$ with $s$, an action $a \in \mathcal{A}$ with $a$ and a reward $r \in [r_{min}, r_{max}]$ with $r$. Given a state-action tuple

$(s, a)$, the probability of the next random state $S_{t+1}$ being $s'$ depends on a *state-transition* distribution: $\mathbb{P}_\mu(S_{t+1} = s' | S_t = s, A_t = a) = \mu(s'|s, a), \forall s, s' \in \mathcal{S}$. We refer to $S_t$ as the *random state* at time $t$. The probability of the action $a$ depends on the agent's policy, which is a stationary mapping $\pi : \mathcal{S} \to \Delta(\mathcal{A})$, from a state to a probability distribution over actions.

These settings give rise to a discrete-time, stateless (Markovian), Random Process (RP) with the additional notions of *actions* to represent decisions and *rewards* for a feedback signal. Given an initial state distribution $\mathbb{P}_{\mu_0}(S_0)$, the process begins with a random state $s_0 \sim \mathbb{P}_{\mu_0}$. Starting from $s_0$, at each time $t$ the agent interacts with the environment by choosing an action $A_t \sim \mathbb{P}_\pi(\cdot|s_t)$, observing the reward $r_t \sim R_t(S_t, A_t)$ and the next state $s_{t+1} \sim \mathbb{P}_\mu$. If a state $s_t$ is also an *absorbing* state ($s \in \overline{\mathcal{S}} \subset \mathcal{S}$), the MDP transitions to the same state $s_t$ with probability 1 and reward 0, and we say that the episode terminates. We refer to the union of each temporal *transition* $(s_t, a_t, r_t, s_{t+1})$ as a *trajectory* or *episode* $d = \{s_t, a_t, r_t, : 0 \le t \le T\}$, where $T$ is the *horizon* of the episode.

We mostly consider episodic settings where the probability of ending in an absorbing state in finite time is 1, resulting in the random horizon $T$. We consider discrete action spaces $\mathcal{A} = \{a_i : 1 \le i \le n\}$ only.

A trajectory is also a random variable in the space of all trajectories $\mathcal{D} = (\mathcal{S} \times \mathcal{A} \times \mathcal{R})^T$, and its distribution is the joint of all of its components $\mathbb{P}_D(D) = \mathbb{P}_{A,S,R}(s_0, a_1, r_1, \ldots, s_T)$. Given an MDP $\mathcal{M} = (\mathcal{S}, \mathcal{A}, R, \mu, \gamma)$ and fixing a policy $\pi$ produces a Markov Process (MP) $\mathcal{M}^\pi$ and induces a distribution over trajectory $\mathbb{P}_{\mu,\pi}(D)$.

We refer to the *return* random variable $Z_t$ as the sum of discounted rewards from time $t$ to the end of the episode, $Z_t = \sum_{k=t}^{T} \gamma^{k-t} R(S_k, A_k)$. The *control* objective of an RL problem is to find a policy $\pi^*$ that maximises the expected return,

$$\pi^* \in \operatorname*{argmax}_{\pi} \mathbb{E}_{\mu,\pi} \left[ \sum_{t=0}^{T} \gamma^t R(S_t, A_t) \right] = \mathbb{E}\left[ Z_0 \right]. \tag{1}$$

**Partially-Observable MDPs (POMDPs).** POMDPs are MDPs in which agents do not get to observe a true state of the environment, but only a transformation of it, and are specified with an additional tuple $\langle \mathcal{O}, \mu_O \rangle$, where $\mathcal{O}$ is an observation space, and $\mu_O : \mathcal{S} \to \Delta(\mathcal{O})$ is an observation kernel, that maps the true environment state to observation probabilities. Because transitioning between observations is not Markovian, policies are a mapping from partial *trajectories*, which we denote as *histories*, to actions. Histories are sequences of transitions $h_t = \{O_0\} \cup \{A_k, R_k, O_{k+1} : 0 < k < t-1\} \in (\mathcal{O} \times \mathcal{A} \times \mathcal{R})^t = \mathcal{H}$.

**Generalised Policy Iteration (GPI).** We now introduce the concept of value functions. The *state value function* of a policy $\pi$ is the expected return of the policy from state $s_t$, $v^\pi(s) = \mathbb{E}_{\pi,\mu}[Z_t | S_t = s]$. The action-value function (or Q-function) of a policy $\pi$ is the expected return of the policy from state $s_t$ if the agent takes $a_t$, $q^\pi(s, a) = \mathbb{E}_{\pi,\mu}[Z_t | S_t = s, A_t = a]$. Policy Evaluation (PE) is then the process that maps a policy $\pi$ to its value function. A canonical PE procedure starts from an arbitrary value function $V_0$ and iteratively applies the Bellman operator, $\mathcal{T}$, such that:

$$\hat{v}_{k+1}^\pi(S_t) = \mathcal{T}^\pi[\hat{v}_k^\pi(S_t)] := \mathbb{E}_{\pi,\mu}\left[ R(S_t, A_t) + \gamma \hat{v}_k(S_{t+1}) \right], \tag{2}$$

where $\hat{v}_k$ denotes the value approximation at iteration $k$, $A_t \sim \mathbb{P}_\pi(\cdot|S_t)$, and $S_{t+1} \sim \mathbb{P}_{\mu,\pi}(\cdot|S_t, A_t)$. The Bellman operator is a $\gamma$-contraction in the $\ell_\infty$ and the $\ell_2$ norms, and its fixed point is the value of the policy $\pi$. Hence, successive applications of the Bellman operator improve the prediction accuracy because the current value gets closer to the true value of the policy. We refer to the PE as the *prediction* objective (Sutton & Barto, 2018). Policy improvement maps a policy $\pi$ to an improved policy:

$$\pi_{k+1}(a|S) = \mathcal{G}[\pi_k, S] = \mathbb{1}_{\{a\}}(\operatorname*{argmax}_{u \in \mathcal{A}} [R(S, u) + \gamma v_k(S')]) = \mathbb{1}_{\{a\}}(\operatorname*{argmax}_{u \in \mathcal{A}} [q_k(S, u)]). \tag{3}$$

We refer to GPI as a general method to solve the *control* problem (Sutton & Barto, 2018) deriving from the composition of PE and Policy Improvement (PI). In particular, we refer to the algorithm that alternates an arbitrary number $k$ of PE steps and one PI step as Modified Policy Iteration (MPI) (Puterman & Shin,

1978; Scherrer et al., 2015). For $k = 1$, MPI recovers Value Iteration, while for $k \to +\infty$, it recovers Policy Iteration. For any value of $k \in [1, +\infty)$, and under mild assumptions, MPI converges to an optimal policy (Puterman, 2014).

In Deep RL we parameterise a policy using a neural network with parameters set $\theta$ and denote the distribution over action as $\pi(a|s, \theta)$. We apply the same reasoning for value functions, with parameters set $\phi$, which leads to $v(s, \phi)$ and $q(s, a, \phi)$ for the state and action value functions respectively.

## 4 Quantifying action influences

We start by answering **Q1.**, which aims to address the problem of *what* to measure, when referring to credit. Since Minsky (1961) raised the Credit Assignment Problem (CAP), a multitude of works paraphrased his words:

- "*The problem of how to incorporate knowledge*" and "*given an outcome, how relevant were past decisions?*" (Harutyunyan et al., 2019),

- "*Is concerned with identifying the contribution of past actions on observed future outcomes*" (Arumugam et al., 2021),

- "*The problem of measuring an actions influence on future rewards*" (Mesnard et al., 2021),

- "*An agent must assign credit or blame for the rewards it obtains to past states and actions*" (Chelu et al., 2022),

- "*The challenge of matching observed outcomes in the future to decisions made in the past*" (Venuto et al., 2022),

- "*Given an observed outcome, how much did previous actions contribute to its realization?*" (Ferret, 2022, Chapter 4.1).

These descriptions converge to Minsky's original question and show agreement in the literature on an informal notion of credit. In this introduction, we propose to reflect on the different metrics that exist in the literature to quantify it. We generalise the idea of *action value*, which often only refers to *q*-values, to that of *action influence*, which describes a broader range of metrics used to quantify the credit of an action. While we do not provide a definitive answer on what credit *should* be, we review how different works in the existing RL literature have characterised it. We now start by developing an intuition of the notion of credit.

Consider Figure 1, inspired to both Figure 1 of Harutyunyan et al. (2019) and to the *umbrella* problem in Osband et al. (2020). The action taken at $x_0$ determines the return of the episode by itself. From the point of view of *control*, any policy that always takes $a'$ in $x_0$ (i.e., $\pi^* \in \Pi^* : \pi^*(a'|x_0) = 1$), and then any other action afterwards, is an optimal policy. From the CAP point of view, some optimal actions, namely those after the first one, do not *actually* contribute to optimal returns. Indeed, alternative actions still produce optimal returns and contribute equally to each other to achieve the goal, so their credit is equal. We can see that, in addition to optimality, credit not only identifies optimal actions but informs them of how *necessary* they are to achieve an outcome of interest.

From the example, we can deduce that credit evaluates actions for their potential to influence an outcome. The resulting CAP is the problem of **estimating the influence** of an action over an outcome from experimental data and describes a pure association between them.

**Why solving the CAP?** Action evaluation is a cornerstone of RL. In fact, solving a control problem often involves running a GPI scheme. Here, the influence of an action drives learning, for it suggests a possible direction to improve the policy. For example, the action-value plays that role in Equation (3). It follows that the quality of the measure of influence fundamentally impacts the quality of the policy improvement. Low quality evaluations can lead the policy to diverge from the optimal one, hinder learning, and slow down progress (Sutton & Barto, 2018; van Hasselt et al., 2018). On the contrary, high quality evaluations provide accurate, robust and reliable signals that foster convergence, sample-efficiency and low variance. While

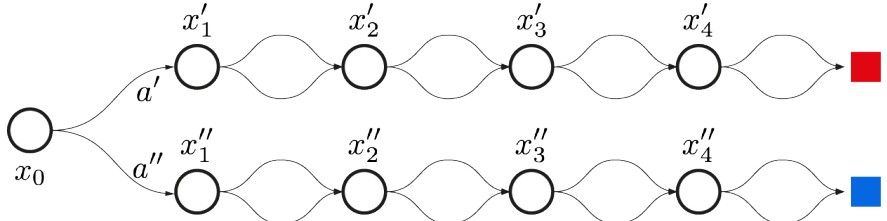

Figure 1: A simplified MDP to develop an intuition of credit. The agent starts in $x_0$, and can choose between two actions, $a'$ and $a''$ in each state; the reward is 1 when reaching the upper, solid red square, and 0 otherwise. The first action determines the outcome alone.

simple evaluations are enough for specialised experiments, the real world is a complex blend of multiple, sometimes hierarchical tasks. In these cases, the optimal value changes from one task to another, and these simple evaluations do not bode well to adapt to general problem solving. Yet, the causal structure that underlies the real word is shared among all tasks, and the modularity of its causal mechanisms is often a valuable property to incorporate. In these conditions, learning to assign credit in one environment becomes a lever to assign credit in another (Ferret et al., 2021a), and ultimately makes learning faster, more accurate and more efficient. For these reasons, and because an optimal policy only requires discovering one single optimal trajectory, credit stores knowledge beyond that expressed by optimal behaviours alone, and solving the control problem is not sufficient to solve the CAP, with the former being an underspecification of the latter.

### 4.1 Are all *action values, credit*?

As we stated earlier, most Deep RL algorithms use some form of *action influence* to evaluate the impacts of an action on an outcome. This is a fundamental requirement to rank actions and select the optimal one to solve complex tasks. For example, many model-free methods use the *state-action value* function $q^\pi(s, a)$ to evaluate actions (Mnih et al., 2015; van Hasselt et al., 2016), where actions contribute as much as the expected return they achieve at termination of the episode. Advantage Learning (AL) (Baird, 1999; Mnih et al., 2016; Wang et al., 2016b, Chapter 5) uses the *advantage* function $A^\pi(s_t, a_t) = q^\pi(s_t, a_t) - v^\pi(s_t)$ [1] to measure credit, while other works study the effects of the *action-gap* (Farahmand, 2011; Bellemare et al., 2016; Vieillard et al., 2020b) on it, that is, the relative difference between the expected return of the best action and that of another action, usually the second best. Action influence is also a key ingredient of actor-critic and policy gradient methods (Lillicrap et al., 2015; Mnih et al., 2016; Wang et al., 2016a), where the policy gradient is proportional to $\mathbb{E}_{\mu,\pi}[A^\pi(s, a)\nabla \log \pi(A|s)]$, with $A^\pi(s, a)$ estimating the influence of the action $A$.

These proxies are sufficient to select optimal actions and unlock solutions to complex tasks (Silver et al., 2018; Wang et al., 2016b; Kapturowski et al., 2019; Badia et al., 2020; Ferret et al., 2021b). However, while many works explicitly refer to the action influence as a measure of credit, the term is not formally defined and, it remains unclear where to draw the line between *credit* and other quantities. Key questions arise: *What is the difference between these quantities and credit? Do they actually represent credit as originally formulated by Minsky (1961)? If so, under what conditions do they do?* Without a clear definition of *what* to measure, we do not have an appropriate quantity to target when designing an algorithm to solve the CAP. More importantly, we do not have an appropriate quantity to use as a single source of truth and term of reference to measure the accuracy of other metrics of action influence, and how well they approximate credit. To fill this gap, we proceed as follows:

- Section 4.2 formalises what is a *goal* or an *outcome*: what we evaluate the action for;

- Section 4.3 unifies existing functions under a common formalism;

---

[1]To be consistent with the RL literature we abuse notation and denote the advantage with a capital letter $A^\pi$ despite not being random and being the same symbol of the action $A_t$.

- Section 4.4 formalises the CAP following this definition;

- Section 4.5 analyses how different works interpreted and quantified *action influences* and reviews them;

- Section 4.6 distils the properties that existing measures of action influence exhibit.

We suggest the reader only interested in the final formalism to directly skip to Section 4.4, and to come back to the next sections to understand the motivation behind it.

## 4.2 What is a *goal*?

Because credit measures the influence of an action upon achieving a certain goal, to define credit formally we must be able to describe *goals* formally, and without a clear understanding of what constitutes one, an agent cannot construct a learning signal to evaluate its actions. *Goal* is a synonym for *purpose*, which we can informally describe as a performance to meet or a prescription to follow. Defining a goal rigorously allows making the relationship between the action and the goal explicit (Ferret, 2022, Chapter 4) and enables the agent to decompose complex behaviour into elementary ones in a compositional (Sutton et al., 1999; Bacon et al., 2017), and possibly hierarchical way (Flet-Berliac, 2019; Pateria et al., 2021; Hafner et al., 2022). This idea is at the foundation of many CA methods (Sutton et al., 1999; 2011; Schaul et al., 2015a; Andrychowicz et al., 2017; Harutyunyan et al., 2019; Bacon et al., 2017; Smith et al., 2018; Riemer et al., 2018; Bagaria & Konidaris, 2019; Harutyunyan et al., 2018; Klissarov & Precup, 2021). We proceed with a formal definition of *goals* in the next paragraph, and review how these goals are *represented* in seminal works on CA in the one after. This will lay the foundation for a unifying notion of credit later in Sections 4.3.

**Defining goals.** To define goals formally, we adopt the *reward hypothesis*, which posits:

> *That all of what we mean by goals and purposes can be well thought of as maximization of the expected value of the cumulative sum of a received scalar signal (reward).* (Sutton, 2004).

Here, the goal is defined as the *behaviour* that results from the process of maximising the return. The reward hypothesis has been further advanced by later studies (Abel et al., 2021b; Pitis, 2019; Shakerinava & Ravanbakhsh, 2022; Bowling et al., 2023). In the following text, we employ the goal definition in Bowling et al. (2023), which we report hereafter:

**Definition 1** (Goal). *Given a distribution of finite histories $\mathbb{P}(H), \forall H \in \mathcal{H}$, we define a* goal *as a partial ordering over $\mathbb{P}(H)$, and for all $h, h' \in \mathcal{H}$ we write $h \succsim h'$ to indicate that $h$ is preferred to $h'$ or that the two are indifferently preferred.*

Here, $H$ is a random history in the set of all histories $\mathcal{H}$ as described in Section 3, and $\mathbb{P}(H)$ is an unknown distribution over histories, different from that induced by the policy and the environment. An agent behaviour and an environment then induce a new distribution over histories, and we obtain $\mathbb{P}_{\mu,\pi}(H)$ as described in Section 3. This in turn allows defining a partial ordering over policies, rather than histories, and we write analogously $\pi \succsim \pi'$ to indicate the preference. For the *Markov Reward Theorem* (Bowling et al., 2023, Theorem 4.1) and under mild conditions (Bowling et al., 2023), there exists a deterministic, Markov reward function[2] $R : \mathcal{O} \times \mathcal{A} \to [0, 1]$ such that the maximisation of the expected sum of rewards is consistent with the preference relation over policies.

**Subjective and objective goals.** The *Markov Reward Theorem* holds both if the preferences are defined *internally* by the agent itself – this is the case of *intrinsic motivation* (Piaget et al., 1952; Chentanez et al., 2004; Barto et al., 2004; Singh et al., 2009; Barto, 2013; Colas et al., 2022) – and in case they originate from an *external* entity, such as an agent-designer. In the first case, the agent doing the maximisation is the

---

[2]We omit the transition dependent discounting for the sake of conciseness and because not relevant to our problem. The reader can consult Pitis (2019); White (2017) for details.

same as the one holding the ordering over policies, and we refer to the corresponding goal as a *subjective goal*. In the second case, an *agent designer* or an unknown, non-observable entity holds the ordering and a separate *learning agent* is the one pursuing the optimisation process. We refer to a goal as an *objective goal* in this latter case. These settings usually correspond to the distinction between goals and sub-goals in the literature (Liu et al., 2022).

**Outcomes.** A particularly interesting use of goals for CA is in hindsight (Andrychowicz et al., 2017). Here the agent acts with a goal in mind, but it evaluates a trajectory as if *a* reward function – one different from the original one – was maximised in the current trajectory. We discuss the benefits of these methods in Section 6.4. When this is the case, we use the term *outcome* to indicate a realised goal in hindsight. In particular, given a history $H \sim \mathbb{P}_{\mu,\pi}(H)$, there exists a deterministic, Markov reward function $R$ that is maximal in $H$. We refer to the corresponding $H$ as an outcome. For example, consider a trajectory $h$ that ends in a certain state $s$. There exist a Markov reward function that outputs always 0 and 1 only when the $s$ is the final state of $h$. We refer to $h$ as an *outcome*.

In other words, this way of defining goals or outcomes corresponds to defining a task to solve, which in turn can be expressed through a reward function with the characteristics described above. Vice-versa, the reward function can *encode* a task. When credit is assigned with respect to a particular goal or outcome, it then evaluates the influence of an action to solving that particular task. As discussed above, this is key to decomposing and recomposing complex behaviours and the definition aligns with that of other disciplines, such as psychology where *a goal . . . is a cognitive representation of something that is possible in the future* (Elliot & Fryer, 2008) or philosophy, where representations do not merely read the world as it is, but they express *preferences* over something that is possible in the future (Hoffman, 2016; Prakash et al., 2021; Le Lan et al., 2022).

**Representing goals and outcomes.** However, expressing the relation between actions and goals explicitly, that is, when the function that returns the credit of an action has a goal as an input, raises the problem of how to *represent* a goal for computational purposes. This is important because among the CA methods that define goals explicitly (Sutton et al., 2011; Schaul et al., 2015a; Andrychowicz et al., 2017; Rauber et al., 2019; Harutyunyan et al., 2019; Tang & Kucukelbir, 2021; Arulkumaran et al., 2022; Chen et al., 2021), not many of them use the rigour of a general-purpose definition of goal such as that in Bowling et al. (2023). In these works, the *goal-representation space*, which we denote as $g \in \mathcal{G}$, is arbitrarily chosen to represent specific features of a trajectory. It denotes an *object*, rather than a performance or a prescription to meet. For example, a *goal-representation g* can be a state (Sutton et al., 2011; Andrychowicz et al., 2017) and $g \in \mathcal{G} = \mathcal{S}$. It can be a specific observation (Nair et al., 2018) with $g \in \mathcal{G} = \mathcal{O}$. Alternatively, it can be an abstract features vector (Mesnard et al., 2021) that reports on some characteristics of a history, and we have $g \in \mathcal{G} = \mathbb{R}^d$, where $d$ is the dimensionality of the vector. Even, a goal can be represented by a natural language instruction (Luketina et al., 2019) and $g \in \mathcal{G} = \mathbb{R}^d$ is the embedding of that piece of text. A goal can be represented by a scalar $g \in \mathcal{G} = \mathbb{R}$ (Chen et al., 2021) that indicates a specific return to achieve, or even a full command (Schmidhuber, 2019), that is a return to achieve is a specific window of time.

While these representations are all useful heuristics, they lack formal rigour and leave space for ambiguities. For example, saying that the goal is a state might mean that *visiting the state at the end of the trajectory* is the goal or that visiting it in the *middle* of it is the goal. This is often not formally defined, and what is the reward function that corresponds to that specific representation of a goal is not always clear. In the following text, when surveying a method or a metric that specifies a goal, we refer to the specific goal representation used in the work and make an effort to detail what is the reward function that underpins that goal representation.

### 4.3 What is an *assignment*?

Having established a formalism for goals and outcomes, we are now ready to describe *credit* formally and we proceed with a formalism that unifies the existing measures of action influence. We first describe a generic function that generalises most CAs, and then proceed to formalise the CAP. Overall, this formulation provides a term of reference for the quantities described in Section 4.5. We now formalise an *assignment*:

**Definition 2** (Assignment). *Consider an action $a \in \mathcal{A}$, a goal $g \in \mathcal{G}$, and a context $c \in \mathcal{C}$. We use the term assignment function or simply assignment to denote a function $\mathcal{K}$ that maps a context, an action, and an outcome to a quantity $y \in \mathcal{Y}$, which we refer to as the **influence** of the action:*

$$K : \mathcal{C} \times \mathcal{A} \times \mathcal{G} \to \mathcal{Y}. \tag{4}$$

Here, a context $c \in \mathcal{C}$ represents some input data and can be arbitrarily chosen depending on the assignment in question. For example, $c$ can be a state $s$. A context must hold information about the present, for example, the current state or the current observation; it may contain information about the past, for example, the sequence of past decisions that occurred until now for a POMDP; to evaluate the current action, it can contain information about what future actions will be taken *in-potentia*, for example by specifying a policy to follow when $a \in \mathcal{A}$ is not taken, or a fixed trajectory, in which case the current action is evaluated in hindsight (Andrychowicz et al., 2017).

In the general case, the action influence is a random variable $Y \in \mathcal{Y} \subset \mathbb{R}^d$. This is the case, for example, of the action-value distribution (Bellemare et al., 2017) as described in Equation 10, where the action influence is defined over the full distribution of returns. However, most methods extract some scalar measures of the full influence distribution, such as expectations (Watkins, 1989), and the action influence becomes a scalar $y \in \mathbb{R}$. In the following text, we mostly consider scalar forms of the influence $\mathcal{Y} = \mathbb{R}$ as these represent the majority of the existing formulations.

In practice, an *assignment* provides a single mathematical form to talk about the multitude of ways to quantify action influence that are used in the literature. It takes an action $a \in \mathcal{A}$, some contextual data $c \in \mathcal{C}$ and a goal $g \in \mathcal{G}$ and maps it to some measure of *action influence*. While maintaining the same mathematical form, different assignments can return different values of action influence and steer the improvement in different directions.

Equation (4) also resembles the General Value Function (GVF) (Sutton et al., 2011), where the influence $y = q^\pi(s, a, g)$ is the expected return of the policy $\pi$ when taking action $a$ in state $s$, with respect a goal $g$. However, in GVFs: *(i)* $y$ is an *action value* and does not generalise other forms of action influence; the goal is an MDP state $g \in \mathcal{S}$ and does not generalise to our notion of goals in Section 4.2; the function only considers forward predictions and does not generalise to evaluating an action in hindsight (Andrychowicz et al., 2017). Table 2 at page 11 contains further details on the comparison and further specifies the relationship between the most common functions and their corresponding assignment.

### 4.4 The credit assignment problem

The generality of the assignment formalism reflects the great heterogeneity of action influence metrics, which we review later in Section 4.5. This heterogeneity shows that, even if most studies agree on an intuitive notion of credit, they diverge in practice in how to quantify credit mathematically. Having unified the existing assignments in the previous section, we now proceed to formalise the CAP analogously. This allows us to put the existing methods into a coherent perspective as a guarantee for a fair comparison, and to maintain the heterogeneity of the existing measures of action influence.

We cast the CAP as the problem of approximating a measure of action influence from experience. We assume standard model-free, Deep RL settings and consider an assignment represented as a neural network $k : \mathcal{C} \times \mathcal{A} \times \mathcal{G} \times \Phi \to \mathbb{R}$ with parameters $\phi \in \Phi = \mathbb{R}^n$ that can be used to approximate the credit of the actions. This usually represents the critic or the value function of an RL algorithm. In addition, we admit a stochastic function to represent the policy, also in the form of a neural network $f : \mathcal{S} \times \Theta \to \Delta(\mathcal{A})$, with parameters set $\theta \in \Theta = \mathbb{R}^m$. We assume that $n \ll |\mathcal{S}| \times |\mathcal{A}|$ and $m \ll |\mathcal{S}| \times |\mathcal{A}|$ and note that often subsets of parameters are shared among the two functions.

We further assume that the agent has access to a set of experiences $\mathcal{D}$ and that it can sample from it according to a distribution $D \sim \mathbb{P}_D$. This can be a pre-compiled set of external demonstrations, where $\mathbb{P}_C(D) = \mathcal{U}(D)$, or an MDP, where $\mathbb{P}_C = \mathbb{P}_{\mu,\pi}(D)$, or even a fictitious model of an MDP $\mathbb{P}_C = \mathbb{P}_{\widetilde{\mu},\pi}(D)$, where $\widetilde{\mu}$ is a function internal to the agent, of the same form of $\mu$. These are also mild assumptions as they

| Assignment | Action influence | Context | Action | Goal |
|---|---|---|---|---|
| State-action-value | $q^\pi(s,a)$ | $s \in \mathcal{S}$ | $a \in \mathcal{A}$ | $g \in \mathbb{R}$ |
| Advantage | $q^\pi(s,a) - v(s)$ | $s \in \mathcal{S}$ | $a \in \mathcal{A}$ | $g \in \mathbb{R}$ |
| General $q$-value function | $q^{\pi,R}(s,a)$ | $s \in \mathcal{S}$ | $a \in \mathcal{A}$ | $g \in \mathcal{S}$ |
| Distributional action-value | $Q^\pi(s,a)$ | $s \in \mathcal{S}$ | $a \in \mathcal{A}$ | $g \in \{0,\dots,n\}$ |
| Distributional advantage | $D_{KL}(Q^\pi(s,a)||V^\pi(s,a))$ | $s \in \mathcal{S}$ | $a \in \mathcal{A}$ | $g \in \{0,\dots,n\}$ |
| Hindsight advantage | $1 - \frac{\pi(A_t|s)}{\mathbb{P}_D(A_t|s_t,Z_t)}Z_t$ | $s \in \mathcal{S}, h_T \in \mathcal{H}$ | $a \in h$ | $g \in \mathbb{R}$ |
| Counterfactual advantage | $\mathbb{P}_D(A_t = a|S_t = s, F_t = f)q(s,a,f)$ | $s \in \mathcal{S}$ | $a \in h$ | $g \in \mathbb{R}$ |
| Posterior value | $\sum_{t=0}^{T}\mathbb{P}_{\mu,\pi}(U_t = u|h_t)v^\pi(o_t,x_t)$ | $o \in \mathcal{O}, u \in \mathbb{R}^d, \pi$ | $A \sim \pi$ | $g \in \mathbb{R}$ |
| Policy-conditioned value | $q(s,a,\pi)$ | $s \in \mathcal{S}, \pi \in \Pi$ | $a \in \mathcal{A}$ | $g \in \mathbb{R}$ |

Table 2: A list of the most common *action influences* and their assignment functions in the Deep RL literature analysed in this survey. For each function, the table specifies the influence, the context representation, the action, and the goal representation of the corresponding assignment function $K \in \mathcal{K}$.

correspond to, respectively, offline settings, online settings, and model-based settings where the model is learned. We detail these settings in Appendix B. We now define the CAP formally.

**Definition 3** (The credit assignment problem). *Consider an MDP $\mathcal{M}$, a goal $g \in \mathcal{G}$, and a set of experience $\mathcal{D}$. Consider an arbitrary assignment $K \in \mathcal{K}$ as described in Equation (4). Given a parameterised function $\widetilde{K} : \mathcal{C} \times \mathcal{A} \times \mathcal{G} \times \Phi \to \mathbb{R}$ with parameters set $\phi \in \Phi \subset \mathcal{R}^n$, we refer to the Credit Assignment Problem as the problem of finding the set of parameters $\phi \in \Phi$ such that:*

$$\widetilde{K}(c,a,g,\phi) = K(c,a,g), \quad \forall c \in \mathcal{C}, a \in \mathcal{A}, g \in \mathcal{G}. \tag{5}$$

Different choices of action influence have a great impact on the hardness of the problem. In particular, there is a trade-off between:

*(a)* how effective the chosen measure of influence is to inform the direction of the policy improvement,

*(b)* how easy it is to learn that function from experience.

For example, using *causal influence* (Janzing et al., 2013) as a measure of action influence makes the CAP hard to solve in practice. In fact, discovering causal mechanisms from associations alone is notoriously challenging (Pearl, 2009; Bareinboim et al., 2022), and pure causal relationships are rarely observed in nature (Pearl et al., 2000) but in specific experimental conditions. However, causal knowledge is reliable, robust to changes in the experience collected and effective, and causal mechanisms can be invariant to changes in the goal. On the contrary, $q$-values are easier to learn as they represent a measure of statistical correlation between state-actions and outcomes, but their knowledge is limited to the bare minimum necessary to solve a control problem. This makes them more brittle to sudden changes to the environment, for example, in open-ended settings (Abel et al., 2023). Which quantity to use in each specific instance or each specific problem is still the subject of investigation in the literature, as we show in the next sections. Ideally, we seek to use the most general measure of influence that can be learned with the least amount of experience.

## 4.5 Existing assignment functions

We now survey the most important assignment functions from the literature and their corresponding measure of action influence. The following list is not exhaustive, but rather it is representative of the limitations of existing credit formalisms. For brevity, and without loss of generality, we omit functions that do not explicitly evaluate actions (for example, state-values), but we notice that it is still possible to reinterpret an assignment to a state as an assignment to a set of actions for it affects all the actions that led to that state.

**State-action values** (Shannon, 1950; Schultz, 1967; Michie, 1963; Watkins, 1989) are a hallmark of RL, and are described by the following expression:

$$q^\pi(s, a) = \mathbb{E}_{\mu,\pi}[Z_t | S_t = s, A_t = a]. \tag{6}$$

Here, the context $c$ is a state $s \in \mathcal{S}$ in the case of MDPs or a history $h \in \mathcal{H}$ for a POMDP. The $q$-function quantifies the credit of an action by the expected return of the action in the context.

$q$-values are among the simplest ways to quantify credit and offer a basic mechanism to solve control problems. However, while $q$-functions offer solid theoretical guarantees in tabular RL, they can be unstable in Deep RL. When paired with bootstrapping and off-policy learning, q-values are well known to diverge from the optimal solution (Sutton & Barto, 2018). van Hasselt et al. (2018) provide empirical evidence of the phenomenon, investigating the relationship between divergence and performance, and how different variables affect divergence. In particular, the work shows that the Deep Q-Network (DQN) (Mnih et al., 2015) is not guaranteed to converge to the optimal $q$-function. The divergence rate on both evaluation and control problems increases depending on specific mechanisms, such as the amount of bootstrapping, or the amount of prioritisation of updates (Schaul et al., 2015b).

An additional problem arises when employing GPI schemes to solve control problems. While during evaluation the policy is fixed, here the policy continuously changes. It becomes more challenging to track the target of the update while converging to it, as the change of policy makes the problem appear non-stationary from the point of view of the value estimation. In fact, even if the policy changes, there is no signal that informs the policy evaluation about the change. To mitigate the issue, many methods either use a fixed network as an evaluation target (Mnih et al., 2015), perform Polyak averaging of the target network (Haarnoja et al., 2018), or clip the gradient update to a maximum cap (Schulman et al., 2017). To further support the idea, theoretical and empirical evidence (Bellemare et al., 2016) shows that the $q$-function is *inconsistent*: for any suboptimal action $a$, the optimal value function $q^*(s, a)$ describes the value of a *non-stationary* policy, which selects a different action $\pi^*(s)$ (rather than $a$) at each visit of $s$.

The non-stationarity of $q$-values for suboptimal actions has also been shown empirically. Schaul et al. (2022) measure the per-state *policy change* $W(\pi, \pi'|s) = \sum_{a \in \mathcal{A}} |\pi(a|s) - \pi'(a|s)|$ for several Atari 2600 games Arcade Learning Environment (ALE) (Bellemare et al., 2013), and show that the action-gap undergoes brutal changes despite the agent maintaining a constant value of expected returns.

In practice, Deep RL algorithms often use $q$-targets to approximate the $q$-value, for example, $n$-step targets (Sutton & Barto, 2018, Chapter 7), or $\lambda$-returns (Watkins, 1989; Jaakkola et al., 1993; Sutton & Barto, 2018, Chapter 12). However, we consider them as *methods*, rather than quantities to measure credit, since they all ultimately aim to converge to the $q$-value. For this reason, we discuss them in Section 6.1.

**Advantage** (Baird, 1999) measures, in a given state, the difference between the q-value of an action and the value of its state

$$A^\pi(s, a) = q^\pi(s, a) - v^\pi(s). \tag{7}$$

Here, the context $c$ is the same as in Equation (6). Because $v^\pi(s) = \sum_{a \in \mathcal{A}} q(s, a)\pi(a|s)$ and $A^\pi(s, a) = q^\pi(s, a) - \mathbb{E}_\pi[q^\pi(s, a)]$, the advantage quantifies how much an action is better than average.

As also shown in Bellemare et al. (2016), using the advantage to quantify credit can increase the *action-gap*. Empirical evidence has shown the consistent benefits of advantage over q-values (Baird, 1999; Wang et al., 2016b; Bellemare et al., 2016; Schulman et al., 2016), and the most likely hypothesis is its regularisation effects (Vieillard et al., 2020b;a; Ferret et al., 2021a). On the other hand, when estimated directly and not by composing state and state-action values, for example in Pan et al. (2022), the advantage does not permit bootstrapping. This is because advantage lacks an absolute measure of action influence, and only maintains one that is relative to the other possible actions.

Overall, in canonical benchmarks for both evaluation (Wang et al., 2016b) and control (Bellemare et al., 2013), advantage has been shown to improve over $q$-values (Wang et al., 2016b). In particular, policy evaluation experiences faster convergence in large action spaces because the state-value $v^\pi(s)$ can hold

information that is shared between multiple actions. For control, it improves the score over several Atari 2600 games compared to both double $q$-learning (van Hasselt et al., 2016) and Prioritised Experience Replay (PER) (Schaul et al., 2015b).

**General Value Functions (GVFs)**   (Sutton et al., 2011; Schaul et al., 2015a) are a set of q-value functions that predict returns for multiple reward functions:

$$q^{\pi,R}(s,a) = \{\mathbb{E}_{\mu,\pi}\left[\sum_t^T R(S_t, A_t)|S_t = s, A_t = a\right] : \forall R \in \mathcal{R}\}, \tag{8}$$

where $R$ is a pseudo-reward function and $\mathcal{R}$ is an arbitrary, pre-defined set of reward functions. Notice that we omit the pseudo-termination and pseudo-discounting terms that appear in their original formulation (Sutton et al., 2011) to maintain the focus on credit assignment. The context $c$ is the same of $q$-values and advantage, and the goal that the pseudo-reward represents is to reach a specific state $g = s \in \mathcal{S}$.

When first introduced (Sutton et al., 2011), the idea of GVFs stemmed from the observation that canonical value functions are limited to address only a single task at a time. Solving a new task would require learning a value function *ex-novo*. By maintaining multiple assignment functions at the same time, one for each goal, GVFs can instantly quantify the influence of an action for multiple goals simultaneously. However, while GVFs maintain multiple assignments, the goal is still not an explicit input of the value function. Instead, it is left implicit, and each assignment serves the ultimate goal to maximise a different pseudo-reward function (Sutton et al., 2011).

Universal Value Functions Approximators (UVFAs) (Schaul et al., 2015a) scale GVFs to Deep RL and advance their idea further by conflating these multiple assignment functions into a single one, represented as a deep neural network. Here, unlike for state-action values and GVFs, the goal is an explicit input of the assignment:

$$q^\pi(s,a,g) = \mathbb{E}_{\mu,\pi}[Z_t|S_t = s, A_t = a, G_t = g]. \tag{9}$$

The action influence here is measured for a goal explicitly. This allows to leverage the generalisation capacity of deep neural networks and to generalise not only over the space of states but also over that of goals.

**Distributional values**   (Jaquette, 1973; Sobel, 1982; White, 1988; Bellemare et al., 2017) consider the full return distribution $Z_t$ instead of its expected value:

$$Q^\pi(s,a) = \mathbb{P}_{\mu,\pi}(Z_t|S_t = s, A_t = a), \tag{10}$$

where $\mathbb{P}_{\mu,\pi}(Z_t)$ is the distribution over returns. Notice that we use uppercase $Q$ to denote the value distribution and the lowercase $q$ for its expectation (Equation (6)).

To translate the idea into a practical algorithm, Bellemare et al. (2017) proposes a discretised version of the value distribution by projecting $\mathbb{P}_{\mu,\pi}(Z_t)$ on a finite support $\mathcal{C} = \{0 \leq i \leq C\}$. The discretised value distribution then becomes $Q^\pi(s,a) = \mathbb{P}_C(Z_t|S_t = s, A_t = a)$, where $\mathbb{P}_C$ is a categorical Bernoulli that describes the probability that a return $c \in \mathcal{C}$ is achieved. Here, the context is the current MDP state and the goal is the expected return. Notice that while the optimal expected value function $q^*(s,a)$ is unique, in general, there are many optimal value distributions since different optimal policies can induce different value distributions.

Experimental evidence (Bellemare et al., 2017) suggests that distributional values provide a better quantification of the action influence, leading to superior results in well known benchmarks for control (Bellemare et al., 2013). However, it is yet not clear why distributional values improve over their expected counterparts. One hypothesis is that predicting for multiple goals works as an auxiliary task (Jaderberg et al., 2017), which often leads to better performance. Another hypothesis is that the distributional Bellman optimality operator proposed in Bellemare et al. (2017) produces a smoother optimisation problem, but the evidence remains weak or inconclusive (Sun et al., 2022).

**Distributional advantage** (Arumugam et al., 2021) proposes a distributional equivalent of the advantage:

$$A^\pi(s,a) = D_{KL}(Q^\pi(s,a)||V^\pi(s)), \tag{11}$$

and borrows the properties of both distributional values and the expected advantage. Intuitively, Equation (11) shows how much knowing the action changes the value distribution. To do so, it measures the change of the value distribution, for a given state-action pair, relative to the distribution for the particular state only. The KL divergence between the two distributions can then be interpreted as the distributional analogue of Equation (7), where the two quantities appear in their expectation instead. The biggest drawback of this measure of action influence is that it is only treated in theory, and there is no empirical evidence that supports distributional advantage as a useful proxy for credit in practice. Future works should consider providing empirical evidence on how this measure of action influence behaves compared to $q$-values and distributional values.

**Hindsight advantage** (Harutyunyan et al., 2019) stems from conditioning the action influence on future states or returns. The return-conditional hindsight advantage function can be written as follows:

$$A^\pi(s,a,z) = \left(1 - \frac{\mathbb{P}_\pi(A_t = a|S_t = s)}{\mathbb{P}_{\mu,\pi}(A_t = a|S_t = s, Z_t = z)}\right)z. \tag{12}$$

Here $A^\pi(s,a,z)$ denotes the return-conditional advantage and $\mathbb{P}_{\mu,\pi}(a_t|S_t = s, Z_t = z)$ is the return-conditional *hindsight distribution* and describes the probability that an action $a$ has been taken in $s$, given that we observed the return $z$ at the end of the episode, after following $\pi$. The context is a state, and the goal is the expected return, which, in this case, corresponds also to the value of the return collected in the current trajectory.

The idea of *hindsight* – initially presented in Andrychowicz et al. (2017) – is that even if the trajectory does not provide useful information for the main goal, it can be revisited as if the goal was the outcome just achieved. Hindsight advantage brings this idea to the extreme and rather than evaluating only for a pre-defined set of goals such as in Andrychowicz et al. (2017), it evaluates for every experienced state or return. Here, the action influence is quantified by that proportion of return determined by the ratio in Equation (12). To develop an intuition of it, if the action $a$ leads to the return $z$ with probability $> 0$ such that $\mathbb{P}_{\mu,\pi}(A_t = a|S_t = s, Z_t = z) > 0$, but the behaviour policy $\pi$ takes $a$ with probability 0, the credit of the action $a$ is 0. There exists also a state-conditional formulation rather than a return-conditional one, and we refer to Harutyunyan et al. (2019) for details on it to keep the description concise.

**Future-conditional advantage** (Mesnard et al., 2021) generalises hindsight advantage to use an arbitrary property of the future:

$$A^\pi(s,a,f) = \mathbb{P}_{\mu,\pi}(A_t = a|S_t = s, F_t = f)q^\pi(s,a,f). \tag{13}$$

Here, $F : \mathcal{D}^T \to \mathbb{R}^n$ is an $n$-dimensional feature of a trajectory $d$, and $F_t$ is that feature for a trajectory that starts at time $t$ and ends at the random horizon $T$. $q^\pi(s,a,f) = \mathbb{E}_{\mu,\pi}[Z_t|S_t = s, F_t = f, A_t = a]$ denotes the future-conditioned state-action value function. The context is a tuple of state and feature $(s, f)$; the goal is the expected return observed at the end of the trajectory. Notice that you can derive the hindsight advantage by setting $F = Z$.

To develop an intuition, $F$ can represent, for example, whether a day is rainy, and the future-conditional advantage expresses the probability of an action $a$, given that the day will be rainy.

**Counterfactual advantage** (Mesnard et al., 2021) proposes a specific choice of $F$ such that $F$ is independent of the current action. This produces a future-conditional advantage that factorises the influence of an action in two components: the contribution deriving from the intervention itself (the action) and the luck represented by all the components not under the control of the agent at the time $t$, such as fortuitous outcomes of the state-transition dynamics, exogenous reward noise, or future actions. The form is the same as that in Equation 13, with the additional condition that the feature $F_t$ is independent of the action $A_t$ and we have $\mathbb{E}_F[D_{KL}(\mathbb{P}(A_t|S_t = s)||\mathbb{P}(A_t|S_t = s, F_t = f))] = 0$.

The main intuition behind *counterfactual advantage* is the following. While to compute counterfactuals we need access to a model of the environment, in model-free settings we can still compute all the relevant information $F_t$ that does not depend on this model. Once learned, a model of $F$ can then represent a valid baseline to compute counterfactuals in a model-free way. To stay in the scope of this section, we detail how to learn this quantity in Section 6.4.

**Posterior value functions** (Nota et al., 2021) reflect on partial-observability and propose a characterisation of the hindsight advantage bespoke to POMDPs. The intuition behind Posterior Value Functions (PVFs) is that the evaluated action only accounts for a small portion of the variance of returns. The majority of the variance is often due to the part of the trajectory that still has to happen. For this reason, incorporating in the baseline information of the future could have a greater impact in reducing the variance of the policy gradient estimator. PVFs focus on the variance of a future-conditional baseline (Mesnard et al., 2021) caused by the partial observability. Nota et al. (2021) factorises a state $s$ into an observable component $o$ and an non-observable one $u$, and formalises the PVF as follows:

$$v_t^\pi(h_t) = \sum_{u \in \mathcal{U}} \mathbb{P}_{\mu,\pi}(U_t = u | h_t) v^\pi(o_t, u_t), \tag{14}$$

where $u \in \mathcal{U}$ is the non-observable component of $s_t$ such that $s = \{u, o\}$. Notice that this method is not taking into account actions. However, it is trivial to derive the corresponding Posterior Action-Value Function (PAVF) as $q^\pi(h_t, a) = R(s_t, a_t) + \gamma v^\pi(h_{t+1})$.

**Policy-conditioned values** (Harb et al., 2020; Faccio et al., 2021) are value functions that include the policy as an input. For example, a policy-conditioned state-action value has the form:

$$q(s, \pi, a) = q^\pi(s, a), \tag{15}$$

but a representation of the policy $\pi$ is used as an explicit input of the influence function. Here, the context is the union of the current MDP state $s$ and the policy $\pi$, and the goal is the expected return at termination.

The main difference with state-action values is that, all else being equal, $q(s, \pi, a, g)$ produces different values *instantly* when $\pi$ varies, since $\pi$ is now an explicit input. For this reason, $q(s, \pi, a)$ can generalise over the space of policies, while $q^\pi(s, a)$ cannot. Using the policy as an input raises the problem of *representing* a policy in a way that can be fed to a neural network. Harb et al. (2020) and Faccio et al. (2021) propose two methods to represent a policy. To keep our attention on the CAP, we refer to their works for further details on possible ways to represent a policy (Harb et al., 2020; Faccio et al., 2021). Here we limit to convey that the problem of representing a policy has been already raised in the literature.

### 4.6 Discussion

The sheer variety of assignment functions described above leads to an equally broad range of metrics to quantify action influence and what is the best assignment function for a specific problem remains an open question. While we do not provide a definitive answer to the question of which properties are necessary or sufficient for an assignment function to output a satisfactory measure of credit, we set out to draw attention to the problem by abstracting out some of the properties that the metrics above share or lack. We identify the following properties of an assignment function and summarise our analysis in Table 3.

**Explicitness.** We use the term *explicitness* when the goal appears as an explicit input of the assignment and it is not left implicit or inferred from experience. Using the goal as an input allows generalising CA over the space of goals. The decision problem can then more easily be broken down into subroutines that are both independent of each other and independently useful to achieve some superior goal $g$.

Overall, explicitness allows incorporating more knowledge because the assignment spans each goal without losing information about others, only limited by the capacity of the function approximator. For example, UVFAs, hindsight advantages, and future conditional advantages are explicit assignments. As discussed in the previous section, *distributional values* can also be interpreted as explicitly assigning credit for each atom

| Name | Explicitness | Recursivity | Future-dependent | Causality |
|---|---|---|---|---|
| State-action value | ○ | ● | ○ | ○ |
| Advantage | ○ | ◐ | ○ | ○ |
| GVFs/UVFAs | ● | ● | ○ | ○ |
| Distributional action-value | ◐ | ● | ○ | ○ |
| Distributional advantage | ◐ | ○ | ○ | ● |
| Hindsight advantage | ◐ | ○ | ◐ | ○ |
| Counterfactual advantage | ◐ | ○ | ◐ | ● |
| Posterior value | ○ | ○ | ● | ○ |
| Observation-action value | ○ | ○ | ○ | ○ |
| Policy-conditioned value | ○ | ● | ● | ○ |

Table 3: A list of the most common *action influences* and their assignment functions in the Deep RL literature analysed in this survey, and the properties they respect. Respectively, empty circles, half circles and bullets indicate that the property is not respected, that it is only partially respected, and it is fully respected. See Sections 4.5 and 4.6 for details.

of the quantised return distribution, which is why we only partially consider them having this property in Table 3. Likewise, hindsight and future-conditional advantage, while not conditioning on a goal explicitly, can be interpreted as conditioning the influence on sub-goals that are states or returns, and future statistics, respectively. For this reason, we consider them as partially explicit assignments.

**Recursivity.** We use the term *recursivity* to characterise the ability of an assignment function to support *bootstrapping* (Sutton & Barto, 2018). When an assignment is recursive, it respects a relationship of the type: $K(c_{t+1}, a_{t+1}, g) = f(K(c_t, a_t, g))$, where $f$ projects the influence from the time $t$ to $t+1$. For example, goal-conditioned $q$-values can be written as: $q^\pi(s_{t+1}, a_{t+1}, g) = R(s_t, a_t, g) + \gamma q^\pi(s_t, a_t, g)$, where $R(s_t, a_t, g)$ is the reward function for the goal $g$.

Recursivity provides key advantages when *learning* credit, which we discuss more in detail in Section 6. In theory, it reduces the variance of the estimation at the cost of a bias (Sutton & Barto, 2018): since the agent does not complete the trajectory, the return it observes is imprecise but varies less. In practice, bootstrapping is often necessary in Deep RL when the length of the episode for certain environments makes full Monte-Carlo estimations intractable due to computational and memory constraints.

When the influence function does not support bootstrapping, the agent must obtain complete episodes to have unbiased samples of the return. For example, Direct Advantage Estimation (DAE) (Pan et al., 2022) uses the advantage function as a measure of credit, but it does not decompose the advantage into its recursive components that support bootstrapping ($q(s, a)$ and $v(s)$), and requires full Monte-Carlo returns to approximate it. This is often ill-advised as it increases the variance of the estimate of the return. For this reason, we consider the advantage to only partially satisfy recursivity.

**Future-dependent.** We use the term *future-dependent* for assignments that take as input information about what actions will be or have been taken *after* the time $t$ at which the action $A_t$ is evaluated. This is key because the influence of the current action depends also on what happens *after* the action. For example, picking up a key is not meaningful if the policy does not lead to opening the door afterwards.

Future actions can be specified *in-potentia*, for example, by specifying a policy to follow after the action. This is the case of policy-conditioned value function, whose benefit is to explicitly condition on the policy such that, if the policy changes, but the action remains the same, the influence of the action changes *instantly*. They can also be specified *in realisation*. This is the case, for example, of hindsight evaluations (Andrychowicz et al., 2017) such as the hindsight advantage, the counterfactual advantage, and the PVF where the influence is conditioned on some features of the future trajectory.

However, these functions only consider *features* of the future: the hindsight advantage considers only the final state or the final return of a trajectory; the counterfactual advantage considers some action-independent

features of the future; the posterior value function considers only the non-observable components. Because futures are not considered fully, we consider these functions as only partially specifying the future.

Furthermore, while state-action value functions, the advantage and their distributional counterparts specify a policy in principle, that information is not an explicit input of the assignment, but only left implicit. In practice, in Deep RL, if the policy changes, the output of these assignments does not change unless retraining.

**Causality.** We refer to a *causal* assignment when the influence that it produces is also a measure of causal influence (Janzing et al., 2013). For example, the counterfactual advantage proposes an interpretation of the action influence closer to causality, by factorising the influence of an action in two. The first factor includes only the non-controllable components of the trajectory (e.g., exogenous reward noise, stochasticity of the state-transition dynamics, stochasticity in the observation kernel), or those not under direct control of the agent at time $t$, such as future actions. The second factor includes only the effects of the action alone. The interpretation is that, while the latter is due to causation, the former is only due to fortuitous correlations. This vicinity to causality theory exists despite the counterfactual advantage not being a satisfactory measure of causal influence as described in Janzing et al. (2013). Distributional advantage in Equation 11 can also be interpreted as containing elements of causality. In fact, we have that the expectation of the advantage over states and actions is the Conditional Mutual Information (CMI) between the policy and the return, conditioned on the state-transition dynamics: $\mathbb{E}_{\mu,\pi}[D_{KL}(Q^\pi(s,a)||V^\pi(s))] = \mathcal{I}(\mathbb{P}_\pi(A|S=s); Z|\mathbb{P}_\mu(S))$. The CMI (with its limitations (Janzing et al., 2013)) is a known measure of causal influence.

Overall, these properties define some characteristics of an assignment, each one bringing positive and negative aspects. Explicitness allows maintaining the influence of an action for multiple goals at the same time, promoting the reuse of information and a compositional onset of behaviour. Recursivity ensures that the influence can be learned via bootstrapping. Future-dependency separates assignments by whether they include information about future actions. Finally, causality filters out the spurious correlations evaluating the effects of the action alone.

### 4.7 Summary

In this section, we addressed **Q1.** and discussed the problem of how to quantify action influences. In Section 4.1 we formalised our question: "*How do different works quantify action influences?*" and "*Are these quantities satisfactory measures of credit?*". We proceeded to answer the questions. In Section 4.2 we formalised the concept of *outcome* as some arbitrary function of a given history. In Section 4.3 we defined the assignment function as a function that returns a measure of action influence. In Section 4.4 we used this definition to formalise the CAP as the problem of learning a measure of action influence from experience. We refer to the set of protocols of this learning process as a credit assignment *method*. In Section 4.5 we surveyed existing measures of action influence from literature, detailed the intuition behind them, their advantages and drawbacks. Finally, in Section 4.6 we discussed how these measures of action influence relate to each other, the properties that they share and those that are rarer in literature, but still promising for future advancements. In the next sections, we proceed to address **Q2.**. Section 5 describes the obstacles to solving the CAP and Section 6 surveys the methods to solve the CAP.

## 5   The challenges to assign credit in Deep RL

Having clarified what measures of action influence are available in the literature, we now look at the obstacles that arise to learn them and, together with Section 6, answer **Q2.**. We first survey the literature to identify *known issues* to assign credit and then systematise the relevant issues into CA challenges. These challenges provide a perspective to understand the principal directions of development of CA methods and are largely independent of the choice of action influence. However, using a measure of influence over another can still impact the prominence of each challenge.

We identify the following issues to assign credit: *(a)* **delayed rewards** (Raposo et al., 2021; Hung et al., 2019; Arjona-Medina et al., 2019; Chelu et al., 2022): reward collection happens long after the action that determined it, causing its influence to be perceived as faint; *(b)* **sparse rewards** (Arjona-Medina et al., 2019;

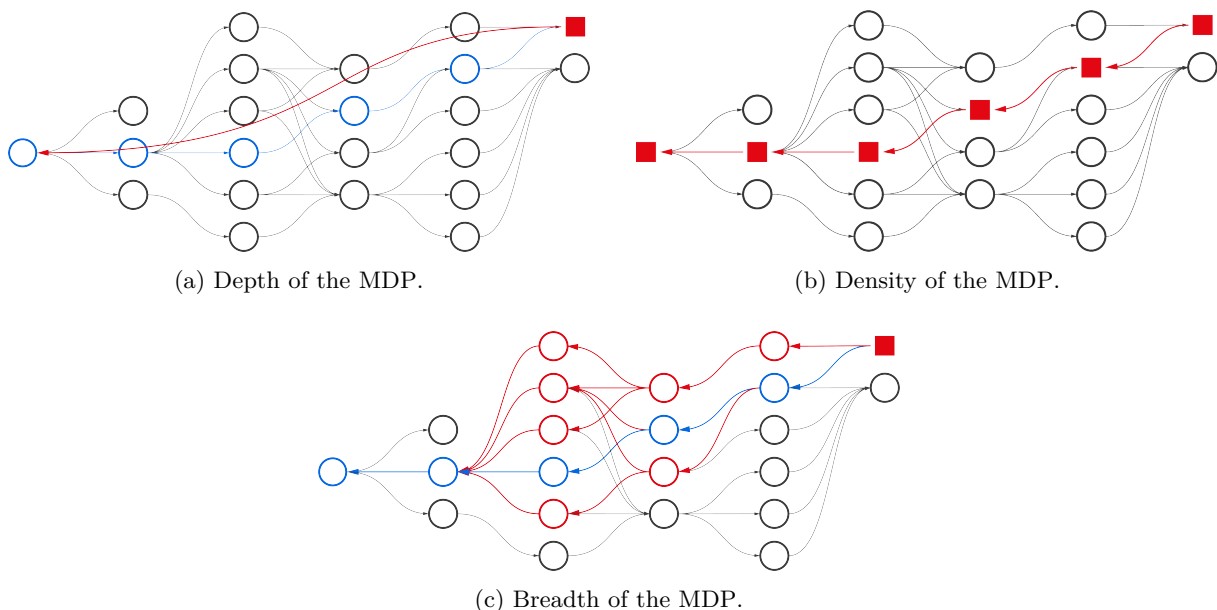

(a) Depth of the MDP.    (b) Density of the MDP.

(c) Breadth of the MDP.

Figure 2: Visual intuition of the three challenges to temporal CA and their respective set of solutions, using the graph analogy. Nodes and arrows represent, respectively, MDP states and actions. Blue nodes and arrows denote the current episode. Black ones show states that could have potentially been visited, but have not. Square nodes denote goals. Forward arrows (pointing right) represent environment interactions, whereas backward arrows (pointing left) denote credit propagation via state-action back-ups. From top left: **(a)** the temporal distance between the accountable action and the target state requires propagating credit deep back in time; **(b)** considering any state as a target increases the density of possible associations and reduces information sparsity; and finally, **(c)** the breadth of possible pathways leading to the target state.

Seo et al., 2019; Chen & Lin, 2020; Chelu et al., 2022): the reward function is zero everywhere, and rarely spikes, causing uninformative Temporal Difference (TD) errors; *(c)* **partial observability** (Harutyunyan et al., 2019): where the agent does not hold perfect information about the current state; *(d)* **high variance** (Harutyunyan et al., 2019; Mesnard et al., 2021; van Hasselt et al., 2021) of the optimisation process; *(e)* the resort to **time as a heuristic** to determine the credit of an action (Harutyunyan et al., 2019; Raposo et al., 2021): *(f)* the lack of **counterfactual** CA (Harutyunyan et al., 2019; Foerster et al., 2018; Mesnard et al., 2021; Buesing et al., 2019; van Hasselt et al., 2021); *(g)* **slow convergence** (Arjona-Medina et al., 2019).

While these issues are all very relevant to the CAP, their classification is also tailored to control problems. Some of these are described by the use of a particular solution, such as *(e)*, or the lack thereof, like *(f)*, rather than by a characteristic of the decision or of the optimisation problem. Here, we systematise these issues and transfer them to the CAP. We identify three principal characteristics of MDPs, which we refer to as *dimensions* of the MDP: **depth**, **density** and **breadth** (see Figure 2). Challenges to CA emerge when pathological conditions on depth, density, and breadth produce specific phenomena that mask the learning signal to be unreliable, inaccurate, or insufficient to correctly reinforce an action. We now detail these three dimensions and the corresponding challenges that arise.

### 5.1 Delayed effects due to high MDP depth

We refer to the *depth* of an MDP as the number of temporal steps that intervene between a highly influential action and an outcome (Ni et al., 2023). When this happens, we refer to the action as a *remote* action, and to the outcome as a *delayed* outcome. When outcomes are delayed, the increase of temporal distance often corresponds to a combinatorial increase of possible alternative futures and the paths to get to them. In these conditions, recognising which action was responsible for the outcome is harder, since the space of

possible associations is very large. We identify two main reasons for an outcome to be delayed, depending on whether the decision after the remote action influences the outcome or not.

The first reason for delayed effects is that the success of the action is not immediate but requires a sequence of actions to be performed *afterwards*, which causes the causal chain leading to success to be long. This issue originates from the typical hierarchical structure of many MDPs, where the agent must first perform a sequence of actions to reach a subjective sub-goal, and then perform another sequence to reach another. The key-to-door task (Hung et al., 2019) is a good example of this phenomenon, where the agent must first collect a key, to be able to open a door later.

The second reason is *delayed reinforcements*: outcomes are only *observed* after a long time horizon, and any decision taken *after* the remote action does not influence the outcome significantly. The phenomenon was first noted in behavioural psychology and is known as the *delayed reinforcement* problem (Lattal, 2010),

> *Reinforcement is delayed whenever there is a period of time between the response producing the reinforcer and its subsequent delivery.* (Lattal, 2010)

The main challenge with *delayed reinforcements* is in being able to ignore the series of irrelevant decisions that are encountered between the remote action and the delayed outcome, focus on the actions that are responsible for the outcome, and assign credit accordingly. This is a key requirement because most CA methods rely on temporal recency as a heuristic to assign credit (Klopf, 1972; Sutton, 1988; Mahmood et al., 2015; Sutton et al., 2016; Jiang et al., 2021a). When this is the case, the actions in the proximity of achieving the goal are reinforced, even if not actually being responsible for the outcome (only the remote action is), just because they are temporally close to the outcome.

While recent works advance proposals on how to measure the MDP depth, for example, CA length (Ni et al., 2023), there is currently no formal agreement in the literature on how to diagnose the presence of delayed effects.

### 5.2 Low action influence due to low MDP density

If delayed effects are characterised by a large temporal distance between an action and the outcome, MDP sparsity derives from a *lack of influence* between them. Even if the literature often confounds *sparse* and *delayed* rewards, there is a substantial difference between them. With delayed effects, actions can cause outcomes very frequently, except with delay. Here, actions have little or no impact on the outcome, and outcomes do not vary regardless of the actions taken, but in a few, rare instances. We identify two main reasons.

The first one is highly stochastic state-transition dynamics, which can be diagnosed by measuring the entropy of the state-transition distribution $\mathcal{H}(\mathbb{P}_\mu)$ and/or of the reward function $\mathcal{H}(\mathbb{P}(R))$. In highly stochastic MDPs, actions hardly affect the future states of the trajectory, the agent is unable to make predictions with high confidence, and therefore cannot select actions that are likely to lead to the goal.

The second reason is the low goal density. This is the canonical case of reward sparsity in RL, where the goal is only achievable in a small subset of the state space, or for a specific sequence of actions. Formally, we can measure the sparsity of an MDP using the notion of information sparsity (Arumugam et al., 2021).

**Definition 4** (MDP sparsity). *An MDP is $\varepsilon$-information sparse if:*

$$\max_{\pi \in \Pi} \mathbb{E}_{\mu,\pi}[D_{KL}(P_{\mu,\pi}(Z|s,a)||P_{\mu,\pi}(Z|s))] \leq \varepsilon, \tag{16}$$

where $\mathbb{E}_{\mu,\pi}$ denotes the expectation over the stationary distribution induced by the policy and the state-transition dynamics. The information sparsity of an MDP is the maximum information gain that can be obtained by an agent. When this is low everywhere, and only concentrated in a small subset of decisions, CA methods often struggle to assign credit, because the probability of behaving optimally is lower (Abel et al., 2021a), and there is rarely a signal to propagate.

### 5.3 Low action influence due to high MDP breadth

We use the term *breadth* of an MDP to denote the number of alternative histories $h$ that produce the same outcome $g$. We then use the term *dilution* of credit, when many optimal pathways exist, and there is no *bottleneck* decision that the agent has to necessarily make to achieve the goal. We formalise the concept using the notion of the *null space* of a policy (Schaul et al., 2022):

$$\text{Null}(\pi) := \{\Omega | v^{\pi}(s) = v^{\pi'}(s)\} \quad \forall \pi, \pi' \in \Omega \subseteq \Pi, \forall s \in \mathcal{S}. \tag{17}$$

$\text{Null}(\pi)$ is the *null space* of a policy $\pi$, defined to be the subset of the space of all policies $\Omega \subset \Pi$ such that two policies $\pi, \pi' \in \Omega$ have the same expected state-value $v^{\pi}(s) = v^{\pi'}(s)$ in all the states of the MDP $s \in \mathcal{S}$.

Credit dilution is often not a challenge for control because optimal behaviours are more probable. However, it can be problematic for CA. Most of the common baselines, such as Advantage Actor Critic (A2C) (Mnih et al., 2016) or Proximal Policy Optimisation (PPO) (Schulman et al., 2017), stop exploring after a small subset of optimal histories is found (or after a certain amount of time). Indeed, when $\text{diam}(\text{Null}(\pi^*))$ is large, there are many optimal histories. Yet, most of them are not included in the experience set $\mathcal{C}$ since exploration stopped prematurely, and credit will not be improved for those. This is particularly relevant for assignments that measure the influence of an action relative to another. For example, the advantage $A^{\pi}(s, a) = q^{\pi}(s, a) - \mathbb{E}_{a' \sim \pi}[q^{\pi}(s, a')]$ is inaccurate if $\mathbb{E}'_a[q^{\pi}(s, a)]$ is inaccurate, which requires taccurately evaluating $q, \forall a' \in \mathcal{A}$. This often results in a low diversity of behaviours (Parker-Holder et al., 2020), and a poor robustness to changes in the environment (Eysenbach & Levine, 2022).

### 5.4 Relationship with the exploration problem

One additional challenge in practical experiments is that it is often hard to disentangle the impacts of CA from those of exploration. In fact, discerning the effects of the two is often only done qualitatively. Here, we discuss the connection between the two problems, if they can be studied independently, and whether it is possible to find a way to diagnose and separate the effect of one from the other.

We use the interpretation of *exploration* as *the problem of acting in an unknown environment to discover temporal sequences of states, actions and rewards with the purpose of acquiring new information* (Amin et al., 2021; Jiang et al., 2023). The acquired experiences then become part of the experience set $\mathcal{C}$, which is used to solve the CAP as described in Equation (5).

To visualise the difference between the exploration problem and the CAP, consider the usual key-to-door environment, where the agent needs to pick up a key, which opens a door, behind which lies a reward. While highly improbable (Abel et al., 2021a), this successful event is the result of chance and random behaviour[3]. Nevertheless, it is the responsibility of *exploration* to *discover* for the **first time** an optimal history, and to keep feeding the set $\mathcal{C}$ with useful discoveries. Then, once the successful experience $\mathcal{C}^*$ is in the set $\mathcal{C}$, it becomes the responsibility of the CA method to consume that experience and extract a measure of influence from the relationship context-action-outcome (Equation (4)) that supports effective improvements.

This is a key difference because the very same behaviour has a different cause whether it comes from exploration or from CA. If due to exploration, it happens by chance, making it unlikely to occur again. If due to accurate CA, it is the result of informed decision-making, and funded on the ability to forecast (Sutton et al., 2011) the effects of an action. Then, when assignments start to be accurate enough, policy improvement further increases the probability of visiting optimal trajectories in a virtuous cycle that also improves CA. Many studies show how common RL baselines often struggle to extract a reliable signal from a small set of isolated successes. This is the case, for example, of A2C (Oh et al., 2018), DQN (Schaul et al., 2015b) or PPO (Arjona-Medina et al., 2019). To further support the claim, increasing the sampling probability of a success, for example through PER (Schaul et al., 2015b) or Self-Imitation Learning (SIL) (Oh et al., 2018), shows great improvements in CA.

We can draw two conclusions from the arguments above. On one hand, if there is a *minimum* number of optimal trajectories $\mathcal{C}^* \subset \mathcal{C}$ in $\mathcal{C}$, exploration has done its job and failures can be attributed to poor CA. On

---

[3]Or, rather, by the laws dictated by the exploration algorithm.

the other hand, a natural question arises: "*What is the minimum rate of successes $G_{min} = |\mathcal{C}^*|/|\mathcal{C}|$ that a CA method requires to start converging to an optimal policy?*". This is a fundamental open question in the current literature, and an answer to it can produce a valid tool to evaluate a CA method. All else being equal, the lowest the ratio $\mathcal{C}^*/\mathcal{C}$, the better the method, because it requires exploration to randomly collect optimal histories at a lower rate, and can solve harder MDPs (Abel et al., 2021a).

## 5.5  Summary

In this section, we surveyed the literature and discussed both the obstacles and the current limitations to solving the CAP. These include delayed rewards, sparse rewards, partial observability, high variance, the lack of counterfactual CA, and sample efficiency. Then, we systematised these issues into challenges that emerge from specific properties of the decision problem, which we refer to as dimensions of the MDP: depth, density, and breadth. Challenges emerge when pathological conditions on these dimensions produce specific phenomena that mask the learning signal to be unreliable, inaccurate, or insufficient to correctly reinforce an action: delayed effects, sparsity, and credit dilution. We have provided an intuition of this classification with the aid of graphs and proceeded to detail each challenge. Finally, we discussed the connection between the CAP and the exploration problem, suggesting a way to diagnose when a failure is caused by one or the other, and disentangling exploration from CA.

With these challenges in mind, we now proceed to review the state of the art in CA, and discuss the methods that have been proposed to address them.

# 6  Methods to assign credit in Deep RL

Following the definition of CAP in Section 4.4, a *credit assignment method* is then an algorithm that takes an initial guess $\widetilde{K}^\phi \in \mathcal{K}$ and a finite set of experience $\mathcal{D} = (\mathcal{S} \times \mathcal{A} \times \mathcal{R})^T$, and, by sampling and learning from transitions $D \sim P_D{}^4$, it recursively produces a better approximation of the true assignment $K$.

In this section, we present a list of the credit assignment methods focused on Deep RL. Our classification aims to identify the principal directions of development and to minimise the intersection between each class of methods. We aim to understand the density around each set of approaches, to locate the branches suggesting the most promising results, and to draw a trend of the latest findings. This can be helpful to both the researchers on the CAP who want to have a bigger picture of the current state of the art, to general RL practitioners and research engineers to identify the most suitable methods to use in their applications, and to the part of the scientific community that focuses on different problems, but that can benefit from the insights on CA. We define a CA method according to how it specifies three elements:

*(a)* The measure of action influence via the assignment function $K$.

*(b)* The protocol that the method uses to approximate $K$ from the experience $\mathcal{D}$.

*(c)* The mechanism $P_D(d)$ to collect and sample from $d \in \mathcal{D}$.

This provides consistency with the framework just proposed and allows categorising each method by the mechanisms that it uses to assign credit. Therefore, for each method, we report the three elements described above. We identify the following categories:

1. Methods using **time contiguity** as a heuristic (Section 6.1).

2. Those **decomposing returns** into per-timestep utilities (Section 6.2).

3. Those conditioning on **predefined goals** explicitly (Section 6.3).

4. Methods conditioning the present on **future outcomes in hindsight** (Section 6.4).

---

[4]To enhance the flow of the manuscript, we formalise *contextual distributions* in Appendix B, and since they are intuitive concepts, we describe them in words when surveying the methods.

5. Modelling trajectories as **sequences** (Section 6.5).

6. Those **planning or learning backwards** from an outcome (Section 6.6).

7. **Meta-learning** different proxies for credit (Section 6.7).

Note that, we do not claim that this list of methods is exhaustive. Rather, as in Section 4.5, this taxonomy is representative of the main approaches and a tool to understand the current state of the art in the field. We are keen to receive feedback on missing methods from the list to improve further revisions of the manuscript. We now proceed to describe the methods, which we also summarise in Table 4.

| Publication | Method | Class | Depth | Density | Breadth |
|---|---|---|---|---|---|
| Baird (1999) | AL | Time | ○ | ○ | ● |
| Wang et al. (2016b) | DDQN | Time | ○ | ○ | ● |
| Pan et al. (2022) | DAE | Time | ○ | ○ | ● |
| Klopf (1972) | ET | Time | ● | ○ | ○ |
| Sutton et al. (2016) | ETD | Time | ● | ○ | ○ |
| Bacon et al. (2017) | Option-critic | Time | ● | ○ | ○ |
| Hung et al. (2019) | TVT | Return decomposition | ● | ○ | ○ |
| Arjona-Medina et al. (2019) | RUDDER | Return decomposition | ● | ○ | ○ |
| Ferret et al. (2021a) | SECRET | Return decomposition | ● | ○ | ○ |
| Ren et al. (2022) | RRD | Return decomposition | ● | ○ | ○ |
| Raposo et al. (2021) | SR | Return decomposition | ● | ○ | ○ |
| Sutton et al. (2011) | GVF | Auxiliary goals | ○ | ● | ○ |
| Schaul et al. (2015a) | UVFA | Auxiliary goals | ○ | ● | ○ |
| Andrychowicz et al. (2017) | HER | Future-conditioning | ○ | ● | ○ |
| Rauber et al. (2019) | HPG | Future-conditioning | ○ | ● | ○ |
| Harutyunyan et al. (2019) | HCA | Future-conditioning | ○ | ● | ○ |
| Schmidhuber (2019) | UDRL | Future-conditioning | ○ | ● | ○ |
| Mesnard et al. (2021) | CCA | Future-conditioning | ○ | ● | ● |
| Nota et al. (2021) | PPG | Future-conditioning | ○ | ● | ○ |
| Janner et al. (2021) | TT | Sequence modelling | ○ | ● | ○ |
| Chen et al. (2021) | DT | Sequence modelling | ○ | ● | ○ |
| Goyal et al. (2019) | Recall traces | Backward planning | ○ | ● | ● |
| Edwards et al. (2018) | FBRL | Backward planning | ○ | ● | ● |
| Nair et al. (2020) | TRASS | Backward planning | ○ | ● | ● |
| van Hasselt et al. (2021) | ET($\lambda$) | Learning predecessors | ● | ○ | ● |
| Xu et al. (2018) | MG | Meta-Learning | ● | ○ | ○ |
| Yin et al. (2023) | Distr. MG | Meta-Learning | ● | ○ | ○ |

Table 4: List of the most representative algorithms for CA classified by the CA challenge they aim to address. For each method, we report the publication that proposed it, the class we assigned to it, and whether it is designed to address each challenge described in Section 5. Hollow circles mean that the method does not address the challenge, and the full circle represents the opposite.

## 6.1 Time as a heuristic

One common way to assign credit is to use time contiguity as a proxy for causality: an action is as influential as it is temporally close to the outcome. This means that, regardless of the action being an actual cause of the outcome, if the action and the outcome appear temporally close in the same trajectory, the action is assigned high credit. At their foundation, there is TD learning (Sutton, 1988), which we describe below.

**TD learning** (Sutton, 1984; 1988; Sutton & Barto, 2018) iteratively updates an initial guess of the value function according to the difference between expected and observed outcomes. More specifically, the agent

starts with an initial guess of values, acts in the environment, observes returns, and aligns the current guess to the observed return. The difference between the expected return and the observed one is the TD error $\delta_t$:

$$\delta_t = R(s_t, a_t) + \gamma q^\pi(s_{t+1}, a_{t+1}) - q^\pi(s_t, a_t) \tag{18}$$

with $a_{t+1} \sim \pi$ and $s_{t+1} \sim \mu$.

When the temporal distance between the goal and the action is high – a premise at the base of the CAP – it is often improbable to observe very far rewards. As time grows, so does the variance of the observed outcome, due to the intrinsic stochasticity of the environment dynamics, and the policy. To mitigate the issue, TD methods often replace the theoretical measure of influence with an approximation: the *TD target*. In TD learning, the value function is updated to approximate the *target*, and not the theoretical measure of action influence underneath it. Since policy improvement uses the current approximation of the value to update the policy, future behaviours are shaped according to it, and the *TD target* drives the learning process.

We separate the methods in this category in three subgroups: those specifically designed around the advantage function, those re-weighing updates to stabilise learning, and those assigning credit to subsets of temporally extended courses of actions.

### 6.1.1 Advantage-based approaches

The first subset of methods uses the *advantage* (see Section 4.5) as a measure of action influence, but still uses time as a heuristic to learn it.

**Actor-Critic (AC)** methods with a baseline function (Sutton & Barto, 2018, Chapter 13) approximate the action influence using some estimator of the *advantage* function (Equation 7). In fact, the policy gradient is proportional to $\mathbb{E}_{\mu,\pi}[(Q^\pi(s,a) - b(s))\nabla \log \pi(a|s)]$ and if we choose $v^\pi(s)$ as our baseline $b(s)$, we get $\mathbb{E}_{\mu,\pi}[(A^\pi(s,a))\nabla \log \pi(a|s)]$ because $q^\pi(s,a) - v^\pi(s,a) = A^\pi(s,a)$. The use of an action-independent baseline function usually helps to reduce the variance of the evaluation, and thus of the policy gradients, while maintaining an unbiased estimate of it (Sutton & Barto, 2018). What function to use as a baseline is the subject of major studies, and different choices of baselines often yield methods that go beyond using time as a heuristic (Harutyunyan et al., 2019; Mesnard et al., 2021; Nota et al., 2021; Mesnard et al., 2023).

**Advantage Learning (AL)** Baird (1999) also uses time as a proxy for causality. There are many instances of AL in the Deep RL literature. The Dueling Deep Q-Network (DDQN) (Wang et al., 2016b) improves on DQN by calculating the q-value as the sum between the state-value function and a normalised version of the advantage. Even if this results in using the q-value as a measure of action influence and $K(s,a) = v^\pi(s) + (A^\pi(s,a) - \sum_a A^\pi(s,a')/|\mathcal{A}|)$, approximating the advantage is a necessary step of it.

DAE (Pan et al., 2022) follows Wang et al. (2016b) with the same specification of the advantage but provides better connections between the advantage and causality theory. In particular, for fully observable MDPs, the causal effect of an action $a$ upon a scalar outcome $G$ is defined as $\mathbb{E}[G|s,a] - \mathbb{E}[G|s]$. If we choose the return $Z$ as outcome, this actually corresponds to the advantage $\mathbb{E}[Z|s,a] - \mathbb{E}[Z|s] = q^\pi(s,a) - v^\pi(s)$, which becomes an approximate expression for the causal influence of an action upon the random return, as discussed also in Arumugam et al. (2021). Here, the context is an MDP state, the action is the greedy action with respect to the current advantage estimation, and the goal is the expected return at termination.

As explained in Section 5.3, advantage can be decomposed in two terms $A^\pi(s,a) = q^\pi(s,a) - v^\pi(s,a)$. Since $v^\pi(s) = \mathbb{E}_\pi[q^\pi(s,a)]$, it is clear that the accuracy of the advantage depends on the accuracy of the $q$-values of all actions. It has been shown that, because of this, estimating and incorporating the advantage in the $q$-value has a regularisation effect (Vieillard et al., 2020a). Another effect is increasing the action-gap (i.e. the difference in value between the best and second-best action), which facilitates value learning. Because evaluations are more accurate for a greater portion of the state-action space, AL-based methods contribute to address MDP breadth, as shown in Table 4.

### 6.1.2 Re-weighing updates and compound targets

The second subset of methods in this category re-weighs temporal updates according to some heuristics, which we detail below. Re-weighing updates can be useful to emphasise or de-emphasise important states or actions to stabilise learning in Deep RL (van Hasselt et al., 2018).

**Eligibility Traces (ET)** (Klopf, 1972; Singh & Sutton, 1996; Precup, 2000a; Geist et al., 2014; Mousavi et al., 2017) credit the long-term impact of actions on future rewards by keeping track of the influence of past actions on the agent's future reward. Specifically, an eligibility trace (Sutton & Barto, 2018, Chapter 12) is a function that assigns a weight to each state-action pair, based on the recency of the last visit to it. A *trace* $e_t(s)$ spikes every time a state (or state-action) is visited and decays exponentially over time until the next visit or until it extinguishes. At each update, the TD error, which determines the magnitude of the update, is scaled by the value of the trace at that state, and $\delta_t^{ET} = \delta_t e_t(s)$. There are several types of eligibility traces, depending on the law of decay of the trace. For example, with accumulating traces (Klopf, 1972), every visit causes an increment of the trace. Replacing traces (Singh & Sutton, 1996) are capped to a specific value, instead.

Deep Q($\lambda$)-Networks (DQ($\lambda$)Ns) (Mousavi et al., 2017) implement eligibility traces on top of DQN (Mnih et al., 2015). Here, the eligibility trace is a vector $e \in \mathbb{R}^d$ with the same number of components $d$ as the parameters of the DNN, and the action influence is measured by the $q$-value with parameters set $\theta \in \mathbb{R}^d$. The context is an MDP state, the action is an off-policy action in a transition arbitrarily chosen from the buffer; the goal is the expected return. The ET information is embedded in the parameters $\theta$ since they are updated according to $\theta \leftarrow \theta + \delta e$. Here $e$ is the eligibility trace, incremented at each update by the value gradient (Sutton & Barto, 2018, Chapter 12): $e \leftarrow \gamma \lambda e + \nabla_\theta q^\pi(s, a)$.

Finally, successive works advanced on the idea of ETs, and proposed different updates for the eligibility vector (Singh & Sutton, 1996; van Hasselt & Sutton, 2015; Precup, 2000a).

**Emphatic Temporal Differences (ETDs)** (Sutton et al., 2016; Mahmood et al., 2015; Jiang et al., 2021b) continue on the idea of ETs to weigh TD updates with a trace. They aim to address the issue that canonical ETs may suffer from early divergence when combined with non-linear function approximation and off-policy learning. The re-weighing in ETD is based on the *emphatic trace*, which encodes the degree of bootstrapping of a state.

Originating from tabular and linear RL, the intuition behind ETDs is that states with high uncertainty – the states encountered long after the state-action pair of evaluation – are more reliable, and vice versa. The main adaptation of the algorithm to Deep RL is by Jiang et al. (2021b), who propose the Windowed Emphatic TD($\lambda$) (WETD) algorithm. In this approach, ETD is adapted to incorporate update windows of length $n$, introducing a mixed update scheme where each state in the window is updated with a variable bootstrapping length, all bootstrapping on the last state in the window. The influence of an action in WETD is the same as for any other ET, but the trace itself is different and measures the amount of bootstrapping of the current estimate.

ETDs provide an additional mechanism to re-weigh updates, the interest function $i : \mathcal{S} \rightarrow [0, \infty)$. By emphasising or de-emphasising the interest of a state, the interest function can be a helpful tool to encode the influence of the actions that had led to that state. Because hand-crafting an interest function requires human interventions, allowing suboptimal and biased results, Klissarov et al. (2022) proposes a method to learn and adapt the interest function at each update using meta-gradients. Improvements on both discrete control, such as ALE, and on continuous control problems, such as MuJoCo (Todorov et al., 2012), suggest that the interest function can be helpful to assign credit faster and more accurately.

Re-weighing updates includes a set of techniques to adjust the influence of past actions based on their temporal proximity to the current state. Such methods aim to mitigate the limitations of TD methods by dynamically adjusting the weight assigned to past actions, thereby emphasizing or de-emphasizing their contribution to future rewards. For this reason, these methods can be seen as potential solutions to mitigate the impacts of delayed effects and improve credit assignment in settings with high MDP depth, as shown in Table 4.

### 6.1.3   Assigning credit to temporally extended actions

The third and last subset of methods in this category assigns credit to temporally extended actions rather than a single, atomic action. This is formalised in the *options framework* (Sutton et al., 1999; Precup, 2000b).

For the purpose of CA, *options*, also known as *skills* (Haarnoja et al., 2017; Eysenbach et al., 2018), can be described as the problem of achieving *sub-goals*, such that an optimal policy can be seen as the composition of elementary behaviours. For example, in a key-to-door environment, such as MiniGrid (Chevalier-Boisvert et al., 2018) or MiniHack (Samvelyan et al., 2021) the agent might select the option *pick up the key*, followed by *open the door*. Each of this macro-action requires a lower level policy to be executed. For example, *pick up the key* requires selecting the actions that lead to reach the key before grabbing it. In the option framework, credit is assigned for each specific subgoal (the macro-action), with the benefits already described for the *explicitness* property, from Section 4.6. The idea stems from the intuition that it is easier to assign credit to macro-actions since a sequence of options is usually shorter than a sequence of atomic actions, reducing the overall temporal distance to the time of achieving the goal.

However, since the option literature often does not explicitly condition on goals, but uses other devices to decompose the CA problem, we review works about learning options next, and dedicate a separate section to auxiliary goal-conditioning in Section 6.3.

**The option-critic architecture**   (Bacon et al., 2017) scales options to Deep RL and mirrors the actor-critic architecture but considering options rather than actions. The option-critic architecture allows learning both how to execute a specific option, and which option to execute at each time simultaneously and online. The option executes using the *call-and-return* model. Starting from a state $s$, the agent picks an option $\omega$ according to its policy over options $\pi_\Omega$. This option then determines the primitive action selection process through the intra-policy $\pi_\omega$ until the option termination function $\beta$ signals to stop. Learning options, and assigning credit to its actions, is then possible using the *intra-option policy gradient* and the *termination gradient* theorems (Bacon et al., 2017), which define the gradient (thus the corresponding update) for all three elements of the learning process: the option $\omega \in \Omega$, their termination function $\beta(s)$ and the policy over options $\pi_\Omega$. Here, the context is a state $s \in \mathcal{S}$, the actions to assign credit to are both the intra-option action $a \in \mathcal{A}$ and the option $\omega \in \Omega$, and the goal is to maximise the return.

On the same lines, Riemer et al. (2018) propose *hierarchical option-critics*, which allows learning options at multiple hierarchical levels of resolution – nested options – but still only on a fixed number of pre-selected options. Klissarov & Precup (2021) further improve on this method by updating all options with a single batch of experience.

In the context of the option-critic architecture, CA occurs at multiple levels of the hierarchy. At the lower, intra-option level, where individual actions are taken, credit assignment involves determining the contribution of each action to the achievement of sub-goals. This is essential for learning effective policies for executing primitive actions within each option. At the higher level of the hierarchy, credit assignment involves attributing credit to options for achieving higher-level goals and involves identifying the contribution of each option to achieving the overall task objective. The hierarchical structure of the option-critic architecture facilitates credit assignment by decomposing the learning problem into multiple levels of abstraction. For their ability to decompose a bigger task into smaller sub-problems, these methods naturally improve credit assignment when effects are delayed and in settings with high MDP depth (see Table 4).

### 6.1.4   Summary and discussion

The methods we covered in this section use the temporal distance between the context-action pair and a reward to measure the action influence. The closer is the action, the higher is its influence and vice versa. While this could maybe be a reasonable assumption when the policy is optimal, it is not the case for the early exploratory stages of learning. In fact, as described in Section 5.4, highly influential actions are often taken long before their rewards are collected while exploring. For example, in our usual key-to-door example, the agent would pick up the key, perform hundreds of random, unnecessary actions, and the goal-tile only reached after those. In these cases, the two events are separated by a "*multitude of* [random and non-influential]

*decisions*" (Minsky, 1961). Because these non-influential actions are temporally closer to reaching the goal-tile than that of picking up the key, these methods mistakenly assign them high influence and, in particular, a higher influence than to pick up the key.

Today, methods that assign credit only by looking at the temporal distance between the action and the outcome usually underperform on tasks with delayed effects (Arjona-Medina et al., 2019). Nevertheless, some of the branches in this category improve assignments in condition of high MDP depth by re-weighing updates, using advantage or breaking down the task into multiple, composable subtasks.

## 6.2 Decomposing return contributions

To improve CA in settings with high MDP depth, the line of research we describe next focuses on decomposing returns into per-timestep contributions. These works interpret the CAP as a *redistribution* problem: the return observed at termination is re-distributed to each time-step with an auxiliary mechanism that depends on each method and complements TD learning.

**Temporal Value Transport (TVT)** (Hung et al., 2019) uses an external long-term memory system to improve on delayed tasks. The memory mechanism is based on the Differentiable Neural Computer (DNC) (Grefenstette et al., 2015; Graves et al., 2016), a neural network then reads events from an external memory matrix, represented as the hidden state of a Long-Short-Term-Memory (LSTM). The agent decides to read from and write into it. To write, state-action-reward triples are projected to a lower dimensional space, and processed by the DNC. During training, this works as a trigger: when a past state-action pair is read from memory, it gets associated with the current one, transporting the state-action value – credit – from the present to the remote state. To read, the state-action-reward is reconstructed from the latent code. During inference, this acts as a proxy for credit. If a past state-action-reward triple is retrieved from the memory, it means that it is correlated with the current return. This allows to use the retrieval score of a past transition as a measure of the influence of its action.

**Return Decomposition for Delayed Rewards (RUDDER)** (Arjona-Medina et al., 2019) stems from the intuition that, if we can construct a reward function that *redistributes* the rewards collected in a trajectory such that the expected future reward is zero, we obtain an instantaneous signal that immediately informs the agent about future rewards. The method proposes to learn a function $f : (\mathcal{S} \times \mathcal{A})^T \to \mathbb{R}$ that maps a sequence of state-action pairs to the sum of discounted rewards, including the past, present and future rewards. In practice, $f$ is implemented as an LSTM, which is trained to fit a subset of the whole experience set $\mathcal{D}_r \subset \mathcal{D}$. $\mathcal{D}_r$ is constructed to contain only trajectories containing delayed rewards, and experience is sampled proportionally to the current prediction error. The underlying hypothesis is that, by fitting the return, the LSTM's hidden state holds useful information to redistribute the return to the most relevant transitions in the sequence.

Once $f$ represents a faithful model of the return, at each iteration of the RL algorithm, RUDDER uses the LSTM to infer the return for each $s_t, a_t \; ind \sim \mathbb{P}_{\mu,\pi}$. It then uses the difference between the inferred returns (i.e., the redistributed returns) at two consecutive time steps as a reward to perform canonical TD learning. This quantity, represents the credit of a state-action pair:

$$K(s_t, a_t) = f(s_{t+1}, a_{t+1}) - f(s_t, a_t) \tag{19}$$

Here, $f(s_{t+1}, a_{t+1}) - f(s_t, a_t) = R^*(s_t, a_t)$ is the reward function of $\mathcal{M}^* = (\mathcal{S}, \mathcal{A}, R^*, \mu, \gamma)$, an MDP return-equivalent to $\mathcal{M} = (\mathcal{S}, \mathcal{A}, R, \mu, \gamma)$: $\mathcal{M}$ and $\mathcal{M}^*$ have the same set of optimal policies, but the reward function $R^*$ of $\mathcal{M}^*$ is such that the sum of expected future rewards is zero for all states (all the future rewards are paid in the current state). The context is a history $h = \{o_t, a_t, r_t : 0 \le t \le T\}$ from the assigned MDP, the action is an action from the trajectory $a \in h$, and the goal is the achieved return.

**Self-Attentional Credit Assignment for Transfer (SECRET)** (Ferret et al., 2021a) uses a causal Transformer-like architecture (Vaswani et al., 2017) with a self-attention mechanism (Lin et al., 2017) in the standalone supervised task of reconstructing the sequence of rewards from observations and actions. It then views attention weights over past state-action pairs as credit for the generated rewards. This was shown to

help in settings of high MDP depth in a way that transfers to novel tasks when trained over a distribution of tasks. We can write its measure of action influence as follows:

$$K(s_t, a_t) = \sum_{t=1}^{T} \mathbb{1}\{S_t = s, A_t = a\} \sum_{i=t}^{T} \alpha_{t \leftarrow i} R(s_i, a_i). \tag{20}$$

Here, $\alpha_{t \leftarrow i}$ is the attention weight on $(o_i, a_i)$ when predicting the reward $r_j$. Also, here the context is a history $h$, the action is an action from the trajectory $a \in h$, and the goal is the achieved return.

**Synthetic returns (SR)** (Raposo et al., 2021) assume only one state-action to be responsible for the terminal reward. They propose a form of state pairs association where the earlier state (the *operant*) is a leading indicator of the reward obtained in the later one (the *reinforcer*). The association model is learned with a form of episodic memory. Each entry in the memory buffer, which holds the states visited in the current episode, is associated with a reward – the *synthetic* reward – via supervised learning. At training time, this allows propagating credit *directly* from the reinforcer to the operant, bypassing the local temporal difference. When this reward model is accurately learned, each time the operant is observed, the synthetic reward model spikes, indicating a creditable state-action pair. Here the synthetic reward acts as a measure of causal influence, and we write:

$$K(s, a) = q^{\pi}(s, a) + f(s). \tag{21}$$

Here $f(s)$ is the synthetic reward function, and it is trained with value regression on the loss $||r_t - u(s_t) \sum_{k=0}^{t-1} f(s_t) - b(s_t)||^2$, where $h(s_t)$ and $b(s_t)$ are auxiliary neural networks optimised together with $f$. As for Arjona-Medina et al. (2019), the context $c$ is a history $h$ from the assigned MDP, the action is an action from the trajectory $a \in h$, and the goal is the achieved return. This method is, however, stable only within a narrow range of hyperparameters and assumes that only one single action is to be credited.

### 6.2.1 Summary and discussion

The methods in this section assign credit by decomposing returns into per time-step contributions and then learning values from this new, clearer reward signal. For the purposes of this survey, they mainly differ by the method used to redistribute the contributions to each context-action pair. TVT uses an external memory system, RUDDER uses contribution analysis, SECRET exploits the Transformer's self-attention weights, SR use a gating function. Their motivation stems from improving on delayed effects, which they often state as an explicit goal, and for this reason, we report them as improving CA in settings of high MDP depth (Table 4).

Indeed, the empirical evidence they provide suggests that improvements are consistent, and *redistribution* methods provide benefits over their TD learning baselines. On the other hand, these methods do not provide formal guarantees that the assignments improve over TD learning, and there is currently a gap to fill to justify these improvements also theoretically. This is the case, for example, of other methods (Harutyunyan et al., 2019; Wang et al., 2016b; Mesnard et al., 2021; van Hasselt et al., 2021) that prove to reduce the variance of the evaluation, some of which we describe in later sections.

### 6.3 Conditioning on a predefined set of auxiliary goals

The methods in this category evaluate actions for their ability to achieve multiple goals explicitly. They do so by conditioning the value function on a goal and then using the resulting value function to evaluate actions. The intuition behind them is that the agent's knowledge about the future can be decomposed into more elementary associations between states and goals. We now describe the two most influential methods in this category.

**General Value Functions (GVFs)** (Sutton et al., 2011), described in Section 4.5, stem from the idea that knowledge about the world can be expressed in the form of predictions. These predictions can then be organised hierarchically to solve more complex problems. While GVFs carry several modifications to the

canonical value, we focus on its goal-conditioning for the purpose of this review, which is also its foundational idea. GVFs conditions the action value on a goal to express the expected return with respect to the reward function that the goal induces. In their original formulation (Sutton et al., 2011), GVFs are a set of value functions, one for each goal. The goal is any object in a predefined goal set of MDP states $g \in \mathcal{S}$, and the resulting measure of action influence is the following:

$$K(s, a, g) = q^\pi(s, a, g), \tag{22}$$

that is the $q$-function with respect to the goal-conditioned reward function $R(s, a, g)$, which is 0 everywhere, and 1 when a certain state is reached. Because GVFs evaluate an action for what it is going to happen in the future, GVFs are forward methods, and interpret the CAP as a prediction problem: "What is the expected return of this action, given that $g$ is the goal?".

**Universal Value Functions Approximators (UVFAs)** (Schaul et al., 2015a) scale the idea of GVFs to a large set of goals, by using a single value function to learn the whole space of goals. One major benefit of UVFAs over GVFs is that they are readily applicable to Deep RL by simply adding the goal as an input to the value function approximator. This allows the agent to learn end-to-end with bootstrapping and allows for exploiting a shared prediction structure across different states and goals. Since they derive from GVFs, UVFA share most of their characteristics. The context is an MDP state $s \in \mathcal{S}$; the goal is still any object in a predefined goal set of states, $g \in \mathcal{S}$, and the credit of an action is the expected return of the reward function induced by the goal (see Equation (22)).

### 6.3.1 Summary and discussion

The methods in this category stand out for using an *explicit* goal to assign credit, as described in Section 4.6. What distinguishes these methods from those that follow in the next section (which also use goals explicitly) is their flexibility. While in hindsight methods choose the goal after completing a trajectory, or based on information acquired during training, these methods do not. Instead, the set of goals of a GVF is predefined in Sutton et al. (2011). UVFAs, even if they can generalise to new goals in theory, they are designed with that purpose in mind, and their application is limited. This represents both a strong limitation of these methods, and a gap to fill in the literature, since it limits both their flexibility and their autonomy to adapt to different tasks, requiring the human designer to specify the set of goals *ex-ante* and to provide the set of goals as an input at the start of training.

Furthermore, their interpretation of credit is still linked to the idea of temporal contiguity described in Section 6.1. For this reason, they share many drawbacks and limitations with those methods and perform poorly when the MDP is deep, especially if not accompanied by more advanced techniques. To the best of our knowledge, there are no examples in the literature that pair these methods with more advanced CA techniques (e.g., *options*), which represents a gap to fill.

On the other hand, by specifying a goal explicitly (GVFs) and by generalising over the goal space (UVFA), conditioning on a predefined set of goals provides a way to extract signals from the environment even when the signal is sparse and action influence is low. In fact, even when the main task is complex, the set of auxiliary goals is designed to provide a useful signal for learning. This is the reason why we consider these methods improving CA when the MDP is sparse (see Table 4).

### 6.4 Conditioning in hindsight

The methods in this category are characterised by the idea of re-evaluating the action influence according to what the agent achieved, rather than what it was supposed to achieve. This means that, given a trajectory $h$, we can choose some goal $g \in \mathcal{G}$ (*after* collecting $h$) and evaluate the influence of all the actions in $h$ upon achieving $g$.

We separate the methods in this category into three subgroups.

  *(i)* Those that re-label the past experience under a different perspective, such as achieving a different goal than the one the agent started.

*(ii)* Those that condition the action evaluation on some properties of the future during training, which becomes an explicit performance request at inference time.

*(iii)* Those that condition on future factors that are independent on the evaluated action, but that still influence future returns.

### 6.4.1 Relabelling experience

**Hindsight Experience Replay (HER)** (Andrychowicz et al., 2017) stems from the problem of learning in sparse rewards environments, which is an example of low action influence in our framework (see Section 5.2. The method exploits the fact that even if a trajectory is suboptimal for the overall implicit goal to maximise MDP returns, it can be viewed as optimal if the goal is to achieve its final state.

In practice, HER brings together UVFAs and experience replay (Lin, 1992) to re-examine trajectories. After collecting a set of trajectories from the environment, the agent stores each transition in a replay buffer, together with both the state it sought to reach and the one that it actually did reach. This allows optimising $\widetilde{K}^\phi(s, a, g)$ for both goals. We refer to this process of re-examining a trajectory collected with a prior goal in mind and evaluating it according to the actually realised outcome as *hindsight conditioning*, which is also the main innovation that HER brings to the CAP. Notice that the original goal is important because the trajectory is collected with a policy that aims to maximise the return for that specific goal.

However, in HER, the goal set is still predefined, which requires additional specifications from the agent-designer and can limit the autonomy of the overall agent, which increases the autonomy of the agent. HER uses the goal-conditioned $q$-values described in Section 6.3 to measure action influence:

$$K(s_t, a_t, s_T) = q^\pi(s_t, a_t, s_T). \tag{23}$$

Here the context is a history from the MDP, the action is an action from the trajectory $a \in h$, and the goal $g$ is to visit a state $s_T$ at the end of a trajectory.

Since HER is limited to off-policy learning with experience replay, **Hindsight Policy Gradients (HPGs)** (Rauber et al., 2019) transfers the findings of HER to Policy Gradient (PG) methods, and extend it to online settings. Instead of updating the policy based on the actual reward received, Hindsight Policy Gradient (HPG) updates the policy based on the hindsight reward, which is calculated based on the new goals that were defined using HER. The main difference with HER is that in HPGs, both the critic and the actor are conditioned on the additional goal. This results in a goal-conditioned policy $\pi(\cdot|S = s, G = g)$, describing the probability of taking an action, given the current state and a realised outcome. The action influence used in HPG is the advantage formulation of the hindsight policy gradients:

$$K(s, a, g) = q^\pi(s, a, g) - v^\pi(s, g), \tag{24}$$

where $q^\pi(s, a, g)$ and $v^\pi(s, g)$ are the goal-conditioned value functions. Here the context $c$ is a history $h = \{o_t, a_t, r_t : 0 \le t \le T\}$, the goal is arbitrarily sampled from a goal set, $g \in \mathcal{G}$. Like HER, HPG is tailored to tasks with low action influence due to low MDP density, and it is shown to be effective in sparse reward settings. Overall, HER and HPG are the first completed work to talk about *hindsight* as the re-examination of outcomes for CA. Their solution is not particularly interesting for the CAP as they do not cast their problem as a CAP and they do not connect the finding to the CAP explicitly. However, they are key precursors of the methods that we review next, which instead provide novel and reusable developments for CAP specifically.

### 6.4.2 Conditioning on the future

**Hindsight Credit Assignment (HCA)** Traditional reinforcement learning algorithms often struggle with credit assignment as they rely solely on foresight: they evaluate actions against a predetermined goal, selected *before* acting. These methods operate under the assumption that we lack knowledge of what occurs beyond a given time step, making accurate credit assignment challenging, especially in tasks with delayed effects. (Harutyunyan et al., 2019), on the other hand, centres on utilising hindsight information, acknowledging that credit assignment and learning typically take place after the agent completes its current trajectory.

This approach enables us to leverage this additional data to refine the learning of critical variables necessary for credit assignment.

(Harutyunyan et al., 2019) introduces a new family of algorithms known as Hindsight Credit Assignment (HCA). Hindsight Credit Assignment (HCA) algorithms explicitly assign credit to past actions based on the likelihood of those actions having been taken, given that a certain outcome has been observed. This is achieved by comparing a learned *hindsight distribution* over actions, conditioned by a future state or return, with the policy that generated the trajectory.

More precisely, the *hindsight distribution*, $h(a|s_t, \pi, g)$ is the likelihood of an action $a$, given the outcome $g$ experienced in the trajectory $d \sim \mathbb{P}_{\mu,\pi}(D|S_0 = s, a_t \sim \pi)$. In practice, Harutyunyan et al. (2019) consider two classes of outcomes: states and returns. We refer to the algorithms that derive from these two classes of goals as *state-HCA* and *return-HCA*. For state-HCA, the context $c$ is the current state $s_t$ at time $t$; the outcome is a future state in the trajectory $s_{t'} \in d$ where $t' > t$; the credit is the ratio between the state-conditional hindsight distribution and the policy $\frac{h_t(a|s_t, s'_t)}{\pi(a|s_t)}$. For return-HCA, the context $c$ is identical; the outcome is the observed return $Z_t$; the credit is the ratio between the return-conditional hindsight distribution and the policy $1 - \frac{\pi(a|s_t)}{h_t(a|s_t, Z_t)}$. The resulting ratios provide a measure of how crucial a particular action was in achieving the outcome. A ratio deviating further from 1 indicates a greater impact (positive or negative) of that action on the outcome. For example, return-HCA measures the influence of an action with the *hindsight advantage* described in Section 4:

$$K(s_t, a_t, z_t) = \left( 1 - \frac{\pi(a_t|S_t = s_t)}{\mathbb{P}_{\mu,\pi}(a_t|S_t = s_t, Z_t = z_t)} \right) z_t. \tag{25}$$

To compute the *hindsight distribution*, HCA algorithms employ a technique related to importance sampling. Importance sampling estimates the expected value of a function under one distribution (the *hindsight distribution*) using samples from another distribution (the policy distribution). In the context of HCA, importance sampling weights are determined based on the likelihood of the agent taking each action in the trajectory, given the hindsight state compared to the likelihood of the policy for that same action. Once the hindsight distribution is computed, HCA algorithms can be used to update the agent's policy and value function. One approach involves using the hindsight distribution to reweight the agent's experience. This means the agent will learn more from actions that were more likely to have contributed to the observed outcome.

Besides advancing the idea of hindsight, (Harutyunyan et al., 2019) carries one novelty: the possibility to drop the typical policy evaluation settings, where the goal is to learn a value function by the repeated application of the Bellman expectation backup. Instead, action values are defined as a measure of the likelihood that the action and the outcome appear together in the trajectory, and are a precursor of the sequence modelling techniques described in the next section (Section 6.5).

**Upside-Down RL (UDRL)** (Schmidhuber, 2019; Srivastava et al., 2019; Ashley et al., 2022; Štrupl et al., 2022) is another implementation of the idea to condition on the future. The intuition behind Upside-Down RL (UDRL) is that rather than conditioning returns on actions, which is the case of the methods in Section 6.1, we can invert the dependency and condition actions on returns instead. This allows using returns as an input and inferring the action distribution that would achieve that return. The action distribution is approximated using a neural network, the *behaviour policy*, that is trained via maximum likelihood estimation using trajectories collected online from the environment. In UDRL the context is a completed trajectory $d$; the outcome is a command that achieves the return $Z_k$ in $H = T - k$ time-steps, which we denote as $g = (Z_k, H)$; the credit of an action $a$ is its probability according to the behaviour function, $\pi(a|s, g)$. In addition to HCA, UDRL also conditions the return to be achieved in a specific timespan.

**Posterior Policy Gradients (PPGs)** (Nota et al., 2021) further the idea of hindsight to provide lower-variance, future-conditioned baselines for policy gradient methods. At the base of PPG there is a novel value estimator, the PVF. The intuition behind PVFs is that in POMDPs the state value is not a valid baseline because the true state is hidden from the agent, and the observation cannot provide as a sufficient statistic for the return. However, after a full episode, the agent has more information to calculate a better, *a*

*posteriori* guess of the state value at earlier states in the trajectory. Nota et al. (2021) refers to the family of possible *a posteriori* estimations of the state value as the PVF. Formally, a PVF decomposes a state into its current observation $o_t$, and some hidden state that is not observable and typically unknown $b_t$. The value of a state can then be written as the expected observation-action value function over the possible non-observable components $u_T \in \mathcal{U} = \mathbb{R}^d$. The action influence of a PPG is quantified by the expression:

$$K(o_t) = \mathop{\mathbb{E}}_{u \in \mathcal{U}} \left[ \mathbb{P}(u_t = u | h_t) v(o_t, u_t) \right]. \tag{26}$$

Notice that, as explained in Section 4.5, the PVF does not depend on an action. However, we can derive the corresponding action-value formulation with $q^\pi(h_t, a) = R(s_t, a_t) + \gamma v^\pi(h_{t+1})$. Here, the context is an observation, the action is the current action and the goal is the observed return. In practice, PVF advances HCA by learning which statistics of the trajectory $\psi(d)$ are useful to assign credit, rather than specifying it objectively as a state or a return.

### 6.4.3 Exposing irrelevant factors

**Counterfactual Credit Assignment (CCA)** For being data efficient, credit assignment methods need to disentangle the effects of a given action of the agent from the effects of external factors and subsequent actions. External factors in reinforcement learning are any factors that affect the state of the environment or the agent's reward but are outside the agent's control. This can include things like the actions of other agents in the environment, changes in the environment state due to natural processes or events. These factors can make credit assignment difficult because they can obscure the relationship between the agent's actions and its rewards.

Mesnard et al. (2021) proposes to get inspiration from the counterfactuals from causality theory to improve credit assignment in model-free reinforcement learning. The key idea is to condition value functions on future events, and learn to extract relevant information from a trajectory. Relevant information here corresponds to all information that is predictive of the return while being independent of the agent's action at time $t$. This allows the agent to separate the effect of its own actions, *the skills*, from the effect of external factors and subsequent actions, the *luck*, which will enable refined credit assignment and therefore faster and more stable learning. It shows that these algorithms have provably lower variance than vanilla policy gradient, and develops valid, practical variants that avoid the potential bias from conditioning on future information. One variant explicitly tries to remove information from the hindsight conditioning that depends on the current action while the second variant avoids the potential bias from conditioning on future information thanks to a technique related to important sampling. The empirical evidence in Mesnard et al. (2021) suggests that CCA offers great improvements in tasks with delayed effects.

### 6.4.4 Summary and discussion

The methods in this section bring many independent novelties to CA. The most relevant for our scope is the idea of hindsight conditioning, which can be summarised as evaluating past actions using additional information about the future, usually not available at the time the action was taken. They differ from those in Section 6.3, as they do not act on a pre-defined objective set of goals, but these are chosen *in hindsight*.

One drawback of these methods is that they must be able to generalise to a large goal space to be effective, which is not a mild requirement because the ability to generalise often correlates with the size of the network. This can limit the applicability of the method, especially in cases of low computation and memory budgets.

One of the greatest benefits of these methods is to always have a signal to learn from because, by construction, there is always a goal that has been achieved in the current trajectory, for example, the final state, or the terminal return. This, in turn, produces a higher number of context-action-outcome associations, translates into additional training data that is often beneficial in supervised problems, and results in an overall denser signal. These improvements in MDPs with low density, which we report in Table 4, are supported by both empirical evidence and theoretical guarantees to reduce the variance of the evaluations (Harutyunyan et al., 2019; Wang et al., 2016b; Mesnard et al., 2021; van Hasselt et al., 2021). Incorporating information about the future (for example, future returns or states), is most likely one major reason why these algorithms

overperform the others. In fact, when this information is designed to express particular features, such as action-independence or independence to irrelevant factors, such as in Mesnard et al. (2021), the gap increases even further.

Finally, some of these methods (Mesnard et al., 2021) also incentivise the discovery of multiple pathways to the same goal, by identifying decisions that are irrelevant to the outcome, resulting in the fact that any of them can be taken without affecting the outcome. The only requirement is to employ an actor-critic algorithm, which we consider a mild assumption, since transitioning from actor-critic to value-based settings is usually trivially achievable.

## 6.5 Modelling trajectories as sequences

The methods in this category are based on the observation that RL can be seen as a sequence modelling problem. Their main idea is to transfer the successes of sequence modelling in Natural Language Processing (NLP) to improve RL.

On a high level, they all share the same assumption: a sequence in RL is a sequence of transitions $(s, a, r)$, and they differ in either how to model the sequence, the problem they solve, or the specific method they transfer from NLP.

**Trajectory Transformers (TTs)** (Janner et al., 2021) implements a decoder-only (Radford et al., 2018; 2019) Transformer (Vaswani et al., 2017) to model the sequence of transitions. TTs learn from an observational stream of data, composed of expert demonstrations resulting in an offline RL training protocol. The main idea of TTs is to model the next token in the sequence, which is composed by the next state, the next action, and the resulting reward. This enables planning, which TTs exploit to plan via beam search. Notice that, for any of these paradigms, if the sequence model is autoregressive – the next prediction depends only on the past history, but since a full episode is available, the future-conditioned probabilities are still well-defined, and also TTs can condition on the future. In TTs the action influence is the product between the action probability according to the demonstration dataset and its $q$-value:

$$K(s_t, a_t, z_t) = \mathbb{P}_\theta(A_t = a_t | Z_t = z_t) q^\pi(s_t, a_t). \tag{27}$$

Here, the context $c$ is an MDP state $c = s \in \mathcal{S}$, the action is arbitrarily selected, and the goal is the return distribution $\mathbb{P}(Z)$.

**Decision Transformers (DTs)** (Chen et al., 2021) proceed on the same lines as TTs but ground the problem in learning, rather than planning. DTs interpret a sequence as a list of $(s_t, a_t, Z_t)$ triples, where $Z_t$ is the return-to-go. They then use a Transformer to learn a model of the actor that takes the current state and the return as input and outputs a distribution over actions. In addition, they optionally learn a model of the critic as well, which takes the current state and each action in the distribution to output the value of each action. The sequences are sampled from expert or semi-expert demonstrations, and the model is trained to maximise the likelihood of the actions taken by the expert. From the perspective of CA, TTs and DTs are equivalent, and they share the same limitation in that they struggle to assign credit accurately to experience beyond that of the offline dataset. Furthermore, like HCA (Harutyunyan et al., 2019), DTs bring more than one novelty to RL. Besides modelling the likelihood of the next token, they also use returns as input to the model, resulting in a form of future conditioning. However, for CA and this section, we are only interested in their idea of sequence modelling and we will not discuss the other novelties. There exist further extensions to DT both to online settings (Zheng et al., 2022) and to model quantities beyond the return (Furuta et al., 2022). The former allows assigning credit by modelling transition sequences in online settings. The latter, instead, generalises sequence modelling to transitions with additional arbitrary information attached – the same way, Future-Conditional Policy Gradient (FC-PG) generalise HCA.

### 6.5.1 Summary and discussion

Sequence modelling in RL transfers the advances in sequence modelling for NLP to Deep RL setting. The main idea is to measure credit by estimating the probability of the next action (or the next token), conditioned

on the context and the goal defined in hindsight, according to an offline dataset of expert trajectories (Chen et al., 2021; Janner et al., 2021).

While some works propose adaptation to online fine-tuning (Lee et al., 2022), these methods mostly learn from offline datasets and the idea to apply sequence modelling online is underexplored. This represents a strong limitation as it limits the generalisation ability of these methods. For example, DT often fail to generalise to returns outside the training distribution.

The distribution that measures this likelihood $\mathbb{P}(a|c, g)$ can be interpreted as the hindsight distribution (Harutyunyan et al., 2018) described in Section 6.4. Their development has a similar pattern to that of hindsight methods and progressively generalises to more complex settings, such as online learning (Zheng et al., 2022) and more general outcomes (Furuta et al., 2022). In practice, these two trends converge together to model the likelihood of action, states and rewards, which hindsight methods call the *hindsight distribution*. Yet, this set of methods would benefit from a better connection to the RL theory. This has been the case for hindsight methods, which leverage notions from causality and the policy gradient theorem (Sutton & Barto, 2018) to achieve better experimental results (Mesnard et al., 2021). For the same reasons explained for hindsight methods in Section 6.4, these methods improve CA when the MDP has low density and the action influence is low (see Table 4).

Nevertheless, sequence modelling remains a promising direction for CA, especially for their ability to scale to large datasets (Reed et al., 2022). It is not clear how these methods position with respect to the CA challenges described in Section 5, for the lack of experimentation on tasks that explicitly stress the agent's ability to assign credit. However, in their vicinity to future-conditioned methods, they bear some of the same advantages and also share some limitations. In particular, for their ability to define outcomes in hindsight, regardless of an objective learning signal, they bode well in tasks with low action influence.

### 6.6 Planning and learning backwards

The methods in this category extend CA to potential predecessor decisions that have not been taken, but could have led to the same outcome (Chelu et al., 2020). The main intuition is that, in environments with low action influence, highly influential actions are rare, and when a goal is achieved the agent should use that event to extract as much information as possible to assign credit to relevant decisions.

We divide the section into two major sub-categories, depending on whether the agent identifies predecessor states by planning with an inverse model, or by learning relevant statistics without it.

### 6.6.1 Planning backwards

**Recall traces** (Goyal et al., 2019) combine model-free updates from Section 6.1 with learning a backward model of the environment. A backward model $\mu^{-1}(s_{t-1}|S_t = s, A_{t-1} = a)$ describes the probability of a state $S_{t-1}$ being the predecessor of another state $s$, given that the action $a$ was taken. This backward action is sampled from a *backward policy*, $\pi_b(a_{t-1}|s_t)$, which predicts the previous action, and a *backward dynamics*.

By autoregressively sampling from the backward policy and dynamics, the agent can cross the MDP backwards, starting from a final state, $s_T$, up until a starting state, $s_0$ to produce a new trajectory, called *recall trace*. This allows the agent to collect experience that always leads to a certain state, $s_T$, but that does so from different starting points, discovering multiple pathways to the same goal.

Formally, the agent alternates between steps of GPI via model-free updates and steps of behaviour cloning on trajectories collected via the backward model. Trajectories are reversed to match the forward arrow of time before cloning. This is a key step towards solving the CAP as it allows propagating credit to decisions that have not been taken but could have led to the same outcome without interacting with the environment directly. Recall-traces measure the influence of an action by its *q*-value, but differ from any other method using the same action influence because the contextual data is produced via backward crossing. The goal is to maximise the expected returns.

The same paradigm has been presented in a concurrent work (Edwards et al., 2018) as Forward-Backward RL (FBRL). The benefits of a backward model have also been further investigated in other studies. Wang et al.

(2021) investigate the problem in offline settings, and show that backward models enable better generalisation than forward ones. van Hasselt et al. (2019) provide empirical evidence suggesting that assigning credit from hypothetical transitions, that is, via planning, improves the overall efficiency in control problems. Chelu et al. (2020) and van Hasselt et al. (2019) further show that backward planning provides even greater benefits than forward planning when the state-transition dynamics are stochastic.

### 6.6.2 Learning predecessors

**Expected Eligibility Trace (ET($\lambda$))**   (van Hasselt et al., 2021) provide a model-free alternative to backward planning that assigns credit to potential predecessors decisions of the outcome: decisions that have been taken in the past but have not in the last episode. The main idea is to weight the action value by its expected eligibility trace, that is, the instantaneous trace (see Section 6.1), but in expectation over the random trajectory, defined by the policy and the state-transition dynamics.

The Deep RL implementation of ET($\lambda$) considers the expected trace upon the action value representation – usually the last layer of a neural network value approximator. Like for other ETs algorithms, ET($\lambda$) measures action influence using the $q$-value of the decision and encodes the information of the trace in the parameters of the function approximator. In this case, the authors interpret the value network as a composition of a non-linear representation function $\phi(s)$ and a linear value function $v(s_t) = w^\top \phi(s)$. The expected trace $e(s) = E\phi(s)$ is then the result of applying a second linear operator $E$ on the representation. $e(s)$ is then trained to minimise the expected $\ell_2$ norm between the current estimation of $e(s)$ and the instantaneous trace.

### 6.6.3 Summary and discussion

The methods in this section assign credit by considering the effects of decisions that have not been taken, but could have led to the same outcome. The intuition behind them is that, in tasks where the action influence is low due to low MDP density, creditable actions are rare findings. When this happens the agent can use that occurrence to extract as much information as possible from them.

One set of methods does so by learning inverse models of the state-transition dynamics and walking backwards from the outcome. Chelu et al. (2020); van Hasselt et al. (2019) further analyse the conditions in which backward planning is beneficial. Another set of methods exploits the idea of eligibility traces and keeps a measure of the marginal state-action probability to assign credit to actions that could have led to the same outcome. Overall, these methods are designed to thrive in tasks where the action influence is low. Also, for their ability to start from a high-value state, backward planning methods can find a higher number of optimal transpositions, and therefore provide a less biased estimate of the credit of a state-action pair.

### 6.7 Meta-learning proxies for credit

The methods in this category aim to meta-learn key hyperparameters of canonical TD methods. In fact, RL methods are often brittle to the choice of hyperparameters, for example, the number of steps to look-ahead in bootstrapping, what discount factor to use, or meta-parameters specific to the method at hand. How to select these meta-parameters is an accurate balance that depends on the task, the algorithm, and the objective of the agent.

For this reason, it is sometimes difficult to analyse them using the usual framework, and we present them differently, by describing their main idea, and the way they are implemented in Deep RL.

**Meta Gradient (MG) RL**   (Xu et al., 2018) remarks how different CA measures of action influence impact the performance on control problems, and proposes to answer the question: "*Among the most common TD targets, which one results in the best performance?*". The method interprets the target as a parametric, differentiable function that can be used and modified by the agent to guide its behaviour to achieve the highest returns.

In particular, *Meta-Gradients* consider the $\lambda$-return (Sutton, 1988) target, for it can generalise the choice of many targets (Schulman et al., 2016). It then learns its meta-parameters: the bootstrapping parameter $\lambda$

and the discount factor $\gamma$. The connection between MG and CA is that, different pairs of meta-parameters evaluate actions differently. For example, changing the discount factor can move the focus of the assignment from early to late actions with effects on policy improvements (Xu et al., 2018). In fact, adapting and learning the meta-parameters online effectively corresponds to meta-learning a measure of action influence, and profoundly affects credit.

Meta-learning credit assignment strategies has been further extended to distributional (Yin et al., 2023) and continual (Zheng et al., 2020) settings. Badia et al. (2020) investigated the effects of meta-learning the discount factor and the exploration rate to balance out short and long-term rewards.

### 6.7.1 Summary and discussion

Overall, these methods assign credit to actions by applying canonical TD learning algorithms with a meta-learnt measure of action influence. The goal can come in the form of an update target (Xu et al., 2018; Zheng et al., 2018; Xu et al., 2020), a full return distribution (Yin et al., 2023), or a reward function (Zheng et al., 2020). This allows agents to adapt their influence function online, especially improving in conditions of high MDP depth.

## 7 Evaluating credit

Like accurate evaluation is fundamental to RL agents to improve their policy, an accurate evaluation of a CA method is fundamental to CA research to monitor if and how a method is advancing the field. The aim of this section is to survey the state of the art of the metrics, the tasks, and the evaluation protocols to evaluate a CA method. We discuss the main components of the evaluation procedure, the performance metrics, the tasks, and the evaluation protocols.

### 7.1 Metrics

We categorise existing metrics to evaluate a CA method in two main classes:

*(a)* The metrics that are already used for control problems. These mostly aim to assess the agent's ability to make optimal decisions, but they do not explicitly measure the accuracy of the action influence.

*(b)* The metrics that target the quality of an assignment directly, which usually aggregate metrics throughout the RL training procedure.

We now proceed to describe the two classes of metrics.

### 7.1.1 Metrics borrowed from control

**Bias, variance and contraction rate.** The first, intuitive, obvious proxy to assess the quality of a credit assignment method is its theoretical performance in suitable *control* problems: the bias, variance, and contraction rate of the policy improvement operator described in Rowland et al. (2020). Notice that these metrics are not formally defined for all the methods, either because some variables cannot be accessed or because the operators they act on are not formally defined for the method in question. For the evaluation operator described in Equation (2), we can specify these quantities as follows.

$$\Gamma = \sup_{s \in \mathcal{S}} \frac{||\mathcal{T}V^\pi(s) - \mathcal{T}V'^\pi(s)||_\infty}{||V^\pi(s) - V'^\pi(s)||_\infty} \tag{28}$$

is the contraction rate and describes how fast the assignment converges to its fixed point, if it does so, and thus how efficient it is. Here $V^\pi(s)$ and $V'^\pi(s)$ are two estimates of the state-value, which highlights that these set of metrics are not suitable to evaluate methods using any measure of action influence.

If $\mathcal{T}$ is contractive, then $\Gamma < 1 \, \forall V^\pi$ and $V'^\pi$, and there exist a fixed-point bias of $\mathcal{T}$ given by:

$$\xi = ||V^\pi(s) - \hat{V}^\pi(s)||_2, \tag{29}$$

where $\hat{V}^\pi(s)$ is the true, unique fixed point of $\mathcal{T}$, whose existence is guaranteed by $\Gamma < 1$. For every evaluation operator $\mathcal{T}$, there is an update rule $\Lambda : \mathbb{R}^{|\mathcal{S}|} \times \mathcal{H} \to \mathbb{R}$ that takes as input the current estimation of the state-value function, and a trajectory and outputs the updated function. $\Lambda$ has a variance:

$$\nu = \mathbb{E}_{\mu,\pi}[||\Lambda[V(s), D] - \mathcal{T}V(s)||_2^2]. \tag{30}$$

These three quantities are usually in a trade-off (Rowland et al., 2020). Indeed, many (if not all) studies on credit assignment (Hung et al., 2019; Mesnard et al., 2021; Ren et al., 2022; Raposo et al., 2021) report the empirical return and its variance. Because the contraction rate is often harder to calculate, an alternative metric is the time-to-performance, which evaluates the number of interactions necessary to reach a given performance. These mostly aim at showing improvement in sample efficiency and/or asymptotic performance. While useful, this is often not enough to assess the quality of credit assignment, as superior returns can be the result of better exploration, better optimisation, better representation learning, luck (as per the environment dynamics' stochasticity) or of a combination of such factors. Using empirical returns makes the evaluation method empirically viable for any measure of action influence described in Section 4, even if these metrics are not formally defined for them. Nonetheless, when the only difference between two RL algorithms lies in how credit is assigned, and this is not confounded by the aforementioned factors, it is generally safe to attribute improvements to superior credit, given that the improvements are statistically significant (Henderson et al., 2018; Agarwal et al., 2021).

**Task completion rate.** A related, but more precise, metric is the success rate. Given a budget of trials, the success rate measures the frequency of task completion, that is, the number of times the task was solved over the total number of episodes: $G = |\mathcal{C}^*|/|\mathcal{C}|$. Here, $\mathcal{C}^*$ is a set of optimal histories experienced by the agent, and $\mathcal{C}$ is the full set of histories used to train it. Considering success rates instead of bias, variance, and trade-off is useful as it alleviates another issue of these performance metrics: there is no distinction between easy-to-optimise rewards and hard-to-optimise rewards. This is evident in the key-to-door task with distractors (Hung et al., 2019), which we describe in detail later in Section 7.2. Due to the stochasticity from the apple phase (the distractors), it is generally impossible to distinguish performance on apple picking (easy-to-optimise rewards) and door opening (hard-to-optimise rewards that superior credit assignment methods usually obtain). Furthermore, the minimum success rate $G_{min}$ could also be an effective metric to disentangle the effects of exploration from those of CA as discussed in Section 5.4, despite never being employed for that purpose. However, notice that this clarity in reporting credit comes at a cost. In fact, even if these kinds of metrics are more precise than performance metrics, they require expert knowledge of the task. They often suffer from the same confounders as bias, variance, and contraction rate.

**Value error.** As the value function is at the heart of many credit assignment methods, another proxy for the quality of the credit is the quality of value estimation, which can be estimated from the distribution of TD errors (Andrychowicz et al., 2017; Rauber et al., 2019; Arjona-Medina et al., 2019). We can then generalise the value error to one of *influence error*: $\mathbb{E}[||\widetilde{K}(s, a, g) - K(s, a, g)||_i]$, where $|| \cdot ||_i$ denotes the $i^{th}$ norm of a vector, $\widetilde{K}(s, a, g)$ is the current approximation of influence and $K(s, a, g)$ is the true influence. A drawback of the influence error (and the value error) is that it can be misleading. When an algorithm does not fully converge, for example, because of high MDP sparsity (see Section *(b)*, it can happen that the value error is very low. This is because the current policy never visits a state with a return different from zero, and the value function collapses to always return zero. Nevertheless, this metric is a viable option to evaluate RL methods that use some form of action influence. It is not applicable, for example, to PG methods using Monte-Carlo returns to improve a parametric policy via gradient ascent (Sutton & Barto, 2018), or to sequence modelling methods (see Section 6.5 that only approximate the action probabilities of a predefined set of demonstrations.

### 7.1.2 Bespoke metrics for credit assignments

We now review metrics that measure the quality of individual credit assignments, that is, how well actions are mapped to corresponding outcomes, or how well outcomes are redistributed to past actions. Usually, these metrics are calculated in hindsight, after outcomes have been observed.

**Using knowledge about the causal structure.** Suppose we have expert knowledge about the causal structure of the task at hand, i.e. which actions cause which outcomes. This is often the case since as humans we often have an instinctive understanding of the tasks agents tackle. In such a case, given an observed outcome from an agent's trajectory, one can compare credit assignments, which approximate such cause and effect relationships, to the ground truth represented by our causal model of the task. We give several examples from the literature. In Delayed Catch, Raposo et al. (2021) assess whether credit is assigned to the actions that lead to catches or to the end-of-episode reward since they know that these actions are causing the experienced rewards. They do the same on the Atari game Skiing, which is a more complex task but that shares the fact that only a subset of the actions of the agent yield rewards. For example, in Skiing, going between ski poles is the only thing that grants rewards (with delay) at the end of an episode. Ferret et al. (2021a) adopt a similar approach and look at the influence attributed to actions responsible for trigger switches in the Triggers environment, which contribute alone to the end-of-episode reward. Similarly, Arjona-Medina et al. (2019) look at redistributions of RUDDER on several tasks, including the Atari 2600 game Bowling.

**Counterfactual simulation.** A natural approach, which is nonetheless seldom explored in the literature, is counterfactual simulation. On a high level, it consists in asking what would have happened if actions that are credited for particular outcomes had been replaced by another action. This is close to the notion of hindsight advantage.

**Comparing to actual values of the estimated quantity.** This only applies to methods whose credit assignments are mathematically grounded, in the sense that they are the empirical approximations of well-defined quantities. In general, one can leverage extra compute and the ability to reset a simulator to arbitrary states to obtain accurate estimations of the underlying quantity, and compare it to the actual, resource-constrained quantity estimated from experience.

### 7.2 Tasks

In what follows, we present environments that we think are most relevant to evaluate credit assignment methods and individual credit assignments. The most significant tasks are those that present all three challenges to assign credit: delayed rewards, transpositions, and sparsity of the influence. This often corresponds to experiments that have reward delay, high marginal entropy of the reward, and partial observability. To benchmark explicit credit assignment methods, we additionally need to be able to recover the ground truth influence of actions w.r.t. given outcomes, or we can use our knowledge of the environment and develop more subjective measures.

### 7.2.1 Diagnostic tasks

Diagnostic tasks are useful as sanity checks for RL agents and present the advantage of running rather quickly, compared to complex environments with visual input that may imply several millions of samples before agents manage to solve the task at hand. Notice that these tasks may not be representative of the performance of the method at scale, but provide a useful signal to diagnose the behaviour of the algorithm in the challenges described in Section 5. Sometimes, the same environment can represent both a diagnostic task and an experiment at scale, simply by changing the space of the observations or the action space.

We first present chain-like environments, that can be represented graphically by a chain (environments **a** to **c**), and then a grid-like environment (environment **d**), that has more natural grid representations for both the environment and the state.

**a) Aliasing chain.** The aliasing chain (introduced in Harutyunyan et al. (2019) as Delayed Effect) is an environment whose outcome depends only on the first action. A series of perceptually aliased and zero-reward states follow this first action, and an outcome is observed at the end of the chain ($+1$ or $-1$ depending on the binary first action).

**b) Discounting chain.** The discounting chain (Osband et al., 2020) is an environment in which a first action leads to a series of states with inconsequential decisions with a final reward that is either $1$ or $1 + \epsilon$, and a variable length. It highlights issues with the discounting horizon.

**c) Ambiguous bandit.** The ambiguous bandit (Harutyunyan et al., 2019) is a variant of a two-armed bandit problem. The agent is given two actions: one that transitions to a state with a slightly more advantageous Gaussian distribution over rewards with probability $1 - \epsilon$, and another that does so with probability $\epsilon$.

**d) Triggers.** Triggers (Ferret et al., 2021a) is a family of environments and corresponding discrete control tasks that are suited for the quantitative analysis of the credit assignment abilities of RL algorithms. Each environment is a bounded square-shaped 2D gridworld where the agent collects rewards that are conditioned on the previous activation of all the triggers of the map. Collecting all triggers turns the negative value of rewards into positive and this knowledge can be exploited to assess proper credit assignment: the actions of collecting triggers appear natural to be credited. The environments are procedurally generated: when requesting a new environment, a random layout is drawn according to the input specifications.

### 7.2.2 Tasks at scale

In the following, we present higher-dimension benchmarks for agents equipped with credit assignment capabilities.

**Atari.** The Arcade Learning Environment (Bellemare et al., 2013) (ALE) is an emulator in which RL agents compete to reach the highest scores on 56 classic Atari games. We list the ones we deem interesting for temporal credit assignment assessment due to delayed rewards, which were first highlighted by Arjona-Medina et al. (2019). **Bowling**: like in real-life bowling, the agent must throw a bowling ball at pins, while ideally curving the ball so that it can clear all pins in one throw. The agent experiences rewards with a high delay, at the end of all rolls (between 2 and 4 depending on the number of strikes achieved). **Venture**: the agent must enter a room, collect a treasure and shoot monsters. Shooting monsters only give rewards after the treasure was collected, and there is no in-game reward for collecting it. **Seaquest**: the agent controls a submarine and must sink enemy submarines. To reach higher scores, the agent has to additionally rescue divers that only provide reward once the submarine lacks oxygen and surfaces to replenish it. **Solaris**: the agent controls a spaceship that earns points by hunting enemy spaceships. These shooting phases are followed by the choice of the next zone to explore on a high-level map, which conditions future rewards. **Skiing**: the agent controls a skier who has to go between poles while going down the slope. The agent gets no reward until reaching the bottom of the slope, at which time it receives a reward proportional to the pairs of poles it went through, which makes for long-term credit assignment.

**VizDoom.** VizDoom (Kempka et al., 2016) is a suite of partially observable 3D tasks based on the classical Doom video game, a first-person shooter. As mentioned before, it is an interesting sandbox for credit assignment because it optionally provides high-level information such as labelled game objects, depth as well as a top-view minimap representation; all of which can be used for approximate optimally efficient credit assignment algorithms.

**BoxWorld.** BoxWorld (Zambaldi et al., 2018) is a family of environments that shares similarities with Triggers, while being more challenging. Environments are also procedurally-generated square-shaped 2D gridworlds with discrete controls. The goal is to reach a gem, which requires going through a series of boxes protected by locks that can only be opened with keys of the same colour while avoiding distractor boxes.

The relations between keys and locks can be utilised to assess assigned credit since the completion of the task (as well as intermediate rewards for opening locks) depends on the collection of the right keys.

**Sokoban.** Sokoban (Racanière et al., 2017) is a family of environments that is similar to the two previous ones. The agent must push boxes to intended positions on the grid while avoiding dead-end situations (for instance, if a block is stuck against walls on two sides, it cannot be moved anymore). While there is no definite criterion to identify decisive actions, actions that lead to dead-ends are known and can be exploited to assess the quality of credit assignment.

**DeepMind Lab.** DeepMind Lab (Beattie et al., 2016) (DMLab) is a suite of partially observable 3D tasks with rich visual input. We identify several tasks that might be of interest to assess credit assignment capabilities, some of which were used in recent work. **Keys-Doors**: the agent navigates to keys that open doors (identified by their shared colour) so that it can get to an absorbing state represented by a cake. Ferret et al. (2021a) consider a harder variant of the task where collecting keys is not directly rewarded anymore and feedback is delayed until opening doors. **Keys-Apples-Doors**: Hung et al. (2019) consider an extended version of the previous task. The agent still has to collect a key, but after a fixed duration a distractor phase begins in which it can only collect small rewards from apples, and finally, the agent must find and open a door with the key it got in the initial phase. To solve the task, the agent has to learn the correlation or causation link between the key and the door, which is made hard because of the extended temporal distance between the two events and of the distractor phase. **Deferred Effects**: the agent navigates between two rooms, the first one of which contains apples that give low rewards, while the other contains cakes that give high rewards but it is entirely in the dark. The agent can turn the light on by reaching the switch in the first room, but it gets an immediate negative reward for it. In the end, the most successful policy is to activate the switch regardless of the immediate cost so that a maximum number of cakes can be collected in the second room before the time limit.

### 7.3 Protocol

**Online evaluation.** The most standard approach is to evaluate the quality of credit assignment methods and individual credit assignments along the RL training procedure. As the policy changes, the credit assignments change since the effect of actions depends on subsequent actions (which are dictated by the policy). One can dynamically track the quality of credit assignments and that of the credit assignment method using the metrics developed in the previous section. For the credit assignment method, since it requires a dataset of interaction, one can consider using the most trajectories produced by the agent. An advantage of this approach is that it allows evaluating the evolution of the credit assignment quality along the RL training, with an evolving policy and resulting dynamics. Also, since the goal of credit assignment is to help turn feedback into improvements, it makes sense to evaluate it in the context of said improvements. While natural, online evaluation means one has little control over the data distribution of the evaluation. This is problematic because it is generally hard to disentangle credit quality from the nature of the trajectories it is evaluated on. A corollary is that outcomes that necessitate precise exploration (which can be the outcomes for which agents would benefit most from accurate credit assignment) might not be explored.

**Offline evaluation.** An alternative is to consider offline evaluation. It requires a dataset of interactions, either collected before or during the RL training. Credit assignments and the credit assignment method then use the parameters learned during the RL training while being evaluated on the offline data. As the policy in the offline data is generally not the latest policy from the online training, offline evaluation is better suited for policy-conditioned credit assignment or (to some extent) trajectory-conditioned credit assignment. Indeed, other forms of credit assignment are specific to a single policy, and evaluating these on data generated from another policy would not be accurate. An important advantage of offline evaluation is that it alleviates the impact of exploration, as one controls the data distribution credit is evaluated on.

# 8 Closing, discussion and open challenges

The CAP is the problem to approximate the influence of an action from a finite amount of experience, and it is of critical importance to deploy RL agents into the real world that are effective, general, safe and interpretable. However, there is a misalignment in the current literature on what credit means in words and how it is formalised. In this survey, we put the basis to reconcile this gap by reviewing the state of the art of the temporal CAP in Deep RL, focusing on three major questions.

## 8.1 Summary

Overall, we observed three major fronts of development around the CAP.

The first concern is the problem of *how to quantify action influence* (**Q1.**). We addressed **Q1.** in **Section 4**, and analysed the quantities that existing works use to represent the influence of an action. In **Section 4.1** we unified these measures of action influence with the *assignment* definition. In Sections 4.3 and 4.5 we showed that the existing literature agrees on an intuition of credit as a measure of the influence of an action over an outcome, but that it does not translate that well into mathematics and none of the current quantities align with the purpose. As a consequence, we proposed a set of principles that we suggest a measure of action influence should respect to represent credit.

The second front aims to address the question of *how to learn action influence from experience* and to describe the existing *methods* to assign credit. In **Section 5** we looked at the challenges that arise from learning these measures of action influence and, together with **Section 6**, answered **Q2.**. We first reviewed the most common obstacles to learning already identified in the literature and realigned them to our newly developed formalism. We identified three dimensions of an MDP, depth, breadth, and density and described pathological conditions on each of them that hinder the CA. In Section 6 we defined a CA method as an algorithm whose aim is to approximate a measure of action influence from a finite amount of experience. We categorised methods into those that: *(i)* use temporal contiguity as a proxy for causal influence; *(ii)* decompose the total return into smaller per-timestep contributions; *(iii)* condition the present on information about the future using the idea of hindsight; *(iv)* use sequence modelling and represent action influence as the likelihood of action to follow a state and predict an outcome; *(v)* learn to imagine backward transitions that always start at a key state and propagate back to the state that could generate them; *(vi)* meta-learn action influence measures.

Finally, the third research front deals with *how to evaluate quantities and methods* to assign credit and aims to provide an unbiased estimation of the progress in the field. In **Section 7** we addressed **Q3.** and analysed how current methods evaluate their performance and how we can monitor future advancements. We discussed the resources that each benchmark has to offer and their limitations. For example, diagnostic benchmarks do not isolate the specific CAP challenges identified in Section 5: delayed effects, transpositions, and sparsity. Benchmarks at scale often cannot disentangle the CAP from the exploration problem, and it becomes hard to understand whether a method is advancing one problem or another.

## 8.2 Discussion and open challenges

As this survey suggests, the work in the field is now fervent and the number of studies in a bullish trend, with many works showing substantial gains in control problems only by – to the best of our current knowledge – advancing on the CAP alone (Bellemare et al., 2017; van Hasselt et al., 2021; Edwards et al., 2018; Mesnard et al., 2021; 2023).

We observed that the take-off of CA research in the broader area of RL research is only recent. The most probable reason for this is to be found in the fact that the tasks considered in earlier Deep RL research were explicitly designed to be simple from the CA point of view. Using tasks where assigning credit is hard would have – and probably still does, e.g., Küttler et al. (2020) – obfuscate other problems that it was necessary to solve before solving the CAP. For example, adding the CAP on the top of scaling RL to high-dimensional observations (Arulkumaran et al., 2017) or dealing with large action spaces (Dulac-Arnold et al., 2015; van Hasselt & Wiering, 2009) would have, most likely, concealed any evidence of progress for the underlying

challenges. This is also why CA methods do not usually shine in classical benchmarks (Bellemare et al., 2013), and peer reviews are often hard on these works. Today, thanks to the advancements in other areas of RL, the field is in a state where improving on the CAP is a compelling challenge.

Yet, the CAP still holds open questions and there is still much discussion required to consider the problem solved. In particular, the following observations describe our positions with respect to this survey.

**Aligning future works to a common problem definition.** The lack of a review since its conception (Minsky, 1961) and the rapid advancements produced a fragmented landscape of definitions for action influence, an ambiguity in the meaning of *credit assignment*, a misalignment between the general intuition and its practical quantification, and a general lack of coherence in the principal directions of the works. While this diversity is beneficial for the diversification of the research, it is also detrimental to comparing the methods. Future works aiming to propose a new CA method should clarify these preliminary concepts. Answers to "What is the choice of the measure of action influence? Why the choice? What is the method of learning it from experience? How is it evaluated?" would be good a starting point.

**Characterising credit.** "*What is the* minimum *set of properties that a measure of action influence should respect to inform control? What the more desirable ones?*". This question remains unanswered, with some ideas in Ferret (2022, Chapter 4), and we still need to understand what characterises a proper measure of credit.

**Causality.** The relationship between CA and causality is underexplored, but in a small subset of works (Mesnard et al., 2021; Pitis et al., 2020; Buesing et al., 2019). The literature lacks a clear and complete formalism that casts the CAP as a problem of causal discovery. Investigating this connection and formalising a measure of action influence that is also a satisfactory measure of causal influence would help better understand the effects of choosing a measure of action influence over another. Overall, we need to better understand the connections between CA and causality: what happens when credit is a strict measure of causal influence? How do current algorithms perform with respect to this measure? Can we devise an algorithm that exploits a causal measure of influence?

**Optimal credit.** Many works refer to *optimal credit* or to *assigning credit optimally*, but it is unclear what that formally means. "*When is credit optimal?*" remains unanswered.

**Combining benefits from different methods.** Methods conditioning on the future currently show superior results compared to methods in other categories. These promising methods include hindsight (Section 6.4), sequence modelling (Section 6.5) and backward learning and planning methods (Section 6.6). However, while hindsight methods are advancing fast, sequence modelling and backward planning methods are underinvestigated. We need a better understanding of the connection between these two worlds, which could potentially lead to even better ways of assigning credit. Could there be a connection between these methods? What are the effects of combining backward planning methods with more satisfactory measures of influence, for example, with CCA?

**Benchmarking.** The benchmarks currently used to review a CA method (Chevalier-Boisvert et al., 2018; Bellemare et al., 2013; Samvelyan et al., 2021) (see Section 7.2) are often borrowed from *control* problems, leading to the issues discussed in Section 7 and recalled in the summary above. On a complementary note, CA methods are often evaluated in actor-critic settings (Harutyunyan et al., 2019; Mesnard et al., 2021), which adds layers of complexity that are not necessary. This, together with the inclusion of other unnecessary accessories, can obfuscate the contributions of CA to the overall RL success. As a consequence, the literature lacks a fair comparison among all the methods, and it is not clear how all the methods in Section 6 behave with respect to each other against the same set of benchmarks. This lack of understanding of the state of the art leads to a poor signal to direct future research. We call for a new, community-driven single set of benchmarks that disentangles the CAP from the exploration problem and isolate the challenges described in Section 5. How to disentangle the CAP and the exploration problem? How to isolate each challenge? Shall we evaluate in value-based settings, and would the ranking between the methods be consistent with an

evaluation in actor-critic settings? While we introduced some ideas in Section 5.4, these questions are still unanswered.

**Reproducibility.**   Many works propose open-source code, but experiments are often not reproducible, their code is hard to read, hard to run and hard to understand. Making code public is not enough, and cannot be considered open-source if it is not easily usable. Other than public, open-source code should be accessible, documented, easy to run, and accompanied by continuous support for questions and issues that may arise from its later usage. We need future research to acquire more rigour in the way to publish, present, and support the code that accompanies scientific publications. In particular, we need *(i)* a formalised, shared and broadly agreed standard that is not necessarily a *new* standard; *(ii)* for new studies to adhere to this standard, and *(iii)* for publishers to review the accompanying code at least as thoroughly as when reviewing scientific manuscripts.

**Monitoring advancements.**   The community lacks a database containing comprehensive, curated results of each baseline. Currently, baselines are often re-run when a new method is proposed. This can potentially lead to comparisons that are unfair both because the baselines could be suboptimal (e.g., in the hyperparameters choice, training regime) and their reproduction could be not faithful (e.g., in translating the mathematics into code). When these conditions are not met, it is not clear whether a new method is advancing the field because it assigns credit better or because of misaligned baselines. We call for a new, community-driven database holding the latest evaluations of each baseline. The evaluation should be driven by the authors and the authors be responsible for its results. When such a database will be available, new publications should be tested against the same benchmarks and not re-run previous baselines, but rather refer to the curated results stored in the database.

**Peer reviewing CA works.**   As a consequence of the issues identified above, and because CA methods do not usually shine in classical benchmarks (Bellemare et al., 2013), peer reviews often do not have the tools to capture the novelties of a method and its improvements. On one hand, we need a clear evaluation protocol, including a shared benchmark and leaderboard to facilitate peer reviews. On the other hand, peer reviews must steer away from using tools and metrics that would be used for control, and use those appropriate for the CAP instead.

**Lack of priors and foundation models.**   Most of the CA methods start to learn credit from scratch, without any prior knowledge but the one held by the initialisation pattern of its underlying network. This represents a main obstacle to making CA efficient because, at each new learning phase, even elementary associations must be learned from scratch. In contrast, when facing a new task, humans often rely on their prior knowledge to determine the influence of an action. In the current state of the art, the use of priors to assign credit more efficiently is overlooked. Vice versa, the relevance of the CAP and the use of more advanced methods for CA (Mesnard et al., 2021; 2023; Edwards et al., 2018; van Hasselt et al., 2021) is often underestimated for the development of foundation models in RL.

### 8.3   Conclusions

To conclude, in this survey, we have set out to formally settle the CAP in Deep RL. The resulting material does not aim to solve the CAP, but rather proposes a unifying framework that enables a fair comparison among the methods that assign credit and organises existing material to expedite the starting stages of new studies. Where the literature lacks answers, we identify the gaps and organise them in a list of challenges. We kindly encourage the research community to join in solving these challenges in a shared effort, and we hope that the material collected in this manuscript can be a helpful resource to both inform future advancements in the field and inspire new applications in the real world.

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
