# A Further related works

The literature also offers surveys on related topics. Liu et al. (2022) review challenges and solutions of Goal-Conditioned Reinforcement Learning (GCRL), and Colas et al. (2022) follow to extend GCRL to Intrinsically Motivated Goal Exploration Process (IMGEP). Both these works are relevant for they generalise RL to multiple goals, but while goal-conditioning is a key ingredient of further arguments (see Section 4.2), GCRL does not aim to address CAP directly. Barto & Mahadevan (2003); Al-Emran (2015); Mendonca et al. (2019); Flet-Berliac (2019); Pateria et al. (2021) survey Hierarchical Reinforcement Learning (HRL). HRL breaks down a long-term task into a hierarchical set of smaller sub-tasks, where each sub-task can be interpreted as an independent goal. However, despite sub-tasks providing intermediate, mid-way feedback that reduces the overall delay of effects that characterises the CAP, these works on HRL are limited to investigate the CAP only by decomposing the problem into smaller ones. Even in these cases, for example in the case of temporally abstract actions (Sutton et al., 1999), sub-tasks either are not always well defined, or they require strong domain knowledge that might hinder generalisation.

# B Further details on contexts

A *contextual distribution* defines a general mechanism to collect the contextual data $c$ (experience). For example, it can be a set of predefined demonstration, an MDP to actively query by interaction, or imaginary rollouts produced by an internal world model. This is a key ingredient of each method, together with its choice of action influence and the protocol to learn that from experience. Two algorithms can use the same action influence measure (e.g., (Klopf, 1972) and (Goyal et al., 2019)), but specify different contextual distributions, resulting in two separate, often very different methods.

Formally, we represent a context as a distribution over some contextual data $C \sim \mathbb{P}_C(C)$, where $C$ is the context, and $\mathbb{P}_C$ is the distribution induced by a specific choice of source. Our main reference for the classification of contextual distributions is the *ladder of causality* (Pearl, 2009; Bareinboim et al., 2022), seeing, doing, imagining, and we define our three classes accordingly.

**Observational distributions** are distributions over a predefined set of data, and we denote it with $\mathbb{P}_{obs}(C)$. Here, the agent has only access to passive set of experience collection from a (possibly unknown) environment. It cannot intervene or affect the environment in any way, but it must learn from the data that is available: it cannot explore. This is the typical case of offline CA methods or methods that learn from demonstrations (Chen et al., 2021), where the context is a fixed dataset of trajectories. The agent can sample from $\mathbb{P}_{obs}$ uniformly at random or with forms of prioritisation (Schaul et al., 2015b; Jiang et al., 2021a). Observational distributions allow assigning credit efficiently and safely since they do not require direct interactions with the environment and can ignore the burden of either waiting for the environment to respond or getting stuck into irreversible states (Grinsztajn et al., 2021). However, they can be limited both in the amount of information they can provide and in the overall coverage of the space of associations between actions and outcomes, often failing to generalise to unobserved associations (Kirk et al., 2023).

**Interactive distributions** are distributions defined by active interactions with an environment, and we denote them with $\mathbb{P}_{\mu,\pi}$. Here, the agent can actively intervene to control the environment through the policy, which defines a distribution over trajectories, $D \sim \mathbb{P}_{\mu,\pi}$. This is the typical case of model-free, online CA methods (Arjona-Medina et al., 2019; Harutyunyan et al., 2019), where the source is the interface of interaction between the agent and the environment. Interactive distributions allow the agent to make informed decisions about which experience to collect (Amin et al., 2021) because the space of associations between actions and outcomes is under the direct control of the agent: they allow exploration. One interesting use of these distributions is to define outcomes in *hindsight*, that is, by unrolling the policy in the environment with a prior objective and then considering a different goal from the resulting trajectory (Andrychowicz et al., 2017). Interactive distributions provide greater information than observational ones but may be more expensive to query, they do not allow to specify all queries, such as starting from a specific state or crossing the MDP backwards, and they might lead to irreversible outcomes with safety concerns (García et al., 2015).

**Hypothetical distributions** are distributions defined by functions internal to the agent, and we denote them with $\mathbb{P}_{\tilde{\mu},\pi}$, where $\tilde{\mu}$ is the agent's internal state-transition dynamic function (learned). They represent potential scenarios, futures or pasts, that do not correspond to actual data collected from the real environment. The agent can query the space of associations surgically and explore a broader space of possible outcomes for a given action without having to interact with the environment. In short, it can imagine a hypothetical scenario, and reason about what would have happened if the agent had taken a different action. Hypothetical distributions enable counterfactual reasoning, that is, to reason about what would have happened if the agent had taken a different action in a given situation. Crucially, they allow navigating the MDP independently of the arrow of time, and, for example, pause the process of generating a trajectory, revert to a previous state, and then continue the trajectory from that point. However, they can produce a paradoxical situation in which the agent explores a region of space with high uncertainty, but relies on a world model that, because of that uncertainty is not very accurate (Guez et al., 2020).

### B.1 Representing a context

Since equation (4) includes a context as an input a natural question arises, "*How to represent contexts?*". Recall that the purpose of the context is to two-fold: *a)*to unambiguously determine the current present as much as possible, and *b)* to convey information about the distribution of actions that will be taken *after* the action we aim to evaluate. Section 4.3 details the reasons of the choice. In many action influence measures (see Section 4.5), such as $q$-values or advantage, the context is only the state of an MDP, or a history if we are solving a POMDP instead. In this case representing the context is the problem of representing a state, which is widely discussed in literature. Notice that this is not about *learning* a state representation, but rather about specifying the shape of a state when constructing and defining an MDP or an observation and an action for a POMDP. These portion of the input addresses the first purpose of a context.

To fulfil its second function the context may contain additional objects and here we discuss only the documented cases, rather than proposing a theoretical generalisation. When the additional input is a policy (Harb et al., 2020; Faccio et al., 2021), then the problem turns to how to represent that specific object. In the specific case of a policy Harb et al. (2020) and Faccio et al. (2021) propose two different methods of representing a policy. In other cases, future actions are specified using a full trajectory, or a feature of it, and in this case the evaluation happens in hindsight. As for policies, the problem turns to representing this additional portion of the context.