# OpenReview forum: "A Survey of Temporal Credit Assignment in Deep Reinforcement Learning"
_TMLR — Accepted by TMLR_

### Review · Reviewer_3PaG · 2024-01-04

**Summary Of Contributions:**

This paper surveys the area of temporal credit assignment in deep reinforcement learning. This paper seeks to organize and compare different types of credit assignment, challenges to credit assignment, methods for learning credit assignment, and credit assignment evaluation procedures from the literature. Toward this end, this paper formalizes notions of action influence and credit assignment, describes desirable properties for an action influence, and organizes contributions by building on these theoretical foundations.

**Audience:**

Yes

**Broader Impact Concerns:**

No broader impact concerns.

**Claims And Evidence:**

Yes

**Requested Changes:**

Please improve Section 6 to explain how each learning method relates or could be combined with each type of assignment described in the previous Section.

Please improve the summaries in Section 6.

Please make the paper more consistent across Sections in terms of the claims that are made.

Please edit the paper to correct typos, improve wording, add missing words, and clarify references.

Some examples in no particular order:

1. At the end of Section 4.1, the last bullet point is missing an "s" and a word. I expect is should read "Section 4.6 distils the properties that existing measure*s* of action influence **exhibit**."
2. There appears to be a hat missing from $V$ in Equation (30).
3. In the summary of Section 6.2, it is unclear to what "their" is referring in the sentence, "Despite their measure of action influence not being a satisfactory quantification of credit . . ."
4. In the last sentence of the summary of Section 6.2, it is unclear to what "these models" refers.
6. In the RUDDER subsection in Section 6.2 just after Equation (20), what is $\mathcal{M}^*$?
7. In the RUDDER subsection, you have $g = z^* \in \mathbb{R}$ at the end of the sentence as if it's describing an optimal policy but I think you meant to describe the maximum expected return. In addition to this confusion, I do not understand why the symbols $g$ and $z^*$ are used here, especially when $g$ is used for the function that returns the discounted sum of rewards along a trajectory, $d$. I also do not understand how $g$ can generate that output without being an explicit function of $d$.
8. Section 5.4 is particularly confusing. The second paragraph of Section 5.4 especially contains many typos and awkward wordings.
9. The first sentence in Section 8.1, "tree" -> "three".

All of these issues should be addressed for me to recommend acceptance.

**Strengths And Weaknesses:**

The first three sections of the paper are an effective setup for a survey on this topic. Section 4 is an interesting discussion and formalization of credit assignment. Section 5 thoughtfully teases apart different types of challenges for effective credit assignment in MDPs.

Unfortunately, Sections 6 and 7 do not make the best use of these foundations. For example, Table 4 contains some interesting claims about the types of challenges defined in Section 5 that different methods address, but it is unclear to me how this table was constructed and I could not find any reference to the table in the text for further discussion.

The paper varies significantly in quality and attention to detail across sections. The paper is largely well written up to Section 4. Section 4 is careful about defining common terms such as "goal", "outcome", "(action) influence", and "(credit) assignment", while the sections that follow are more ambiguous in their language, particularly around the terms "value" and "return". It is often unclear to me if these "values" and "returns" are meant to reference only cumulative or discounted future rewards, and thus only pertain to state value, action value, or advantage credit assignments only or if they are referring to "values" and "returns" in a more general sense.

The method descriptions in Section 6 can be effective in that they describe how a method works and how it relates to previously discussed concepts. For example, the RUDDER subsection in Section 6.2 is effective in this way (in spite of some shortcomings, see items in Requested Changes on this subsection). These method descriptions can also be poor in that they do little more than present a reference and a high level description of the algorithm, without highlighting its relationship to credit assignment. For example, the PGIF subsection in Section 6.4.2.

The summaries at the subsections in Section 6 are confusing. Related, summaries or later parts of the paper may make strong declarative statements that are surprising given the previous content. For example, in the summary of Section 6.2, it states that a measure of action influence is not a satisfactory quantification of credit and cites Section 4.3, but Section 4.3 defines a general notion of assignment that each quantity talked about so far fits within. Another example is in the third paragraph of Section 8.1 where it states that "we highlighted that current benchmarks are not fit for the purpose". While "the purpose" could be clarified, my sense of the discussion of benchmarks in the previous section were that each benchmark had something different to offer, not that all of them were critically lacking.

---

> ### Author Response · Authors · 2024-02-07
> **Official response to Reviewer 3PaG**
>
> Thank you very much for the constructive review and the valuable insights on the manuscript. To address the concerns raised above:
>
> > Please improve Section 6 to explain how each learning method relates or could be combined with each type of assignment described in the previous Section.
> - We have thoroughly improved Section 6 and increased its level of detail. In particular, we ensured that the foundational concepts laid out in the initial sections are consistently applied throughout.
> - We have added a more rigorous definition of a credit assignment method that uses the foundations laid in Section 4.
> - We have added details to the classes of methods about which measure of influence they use and better highlighted pros and cons of the methods.
>
> > Please improve the summaries in Section 6.
>
> > Please make the paper more consistent across Sections in terms of the claims that are made.
> - We have removed the unsupported claims from the summary of Section 6 and checked all the other summaries for consistency, making sure that they contain only claims that are well-supported.
>
> > Please edit the paper to correct typos, improve wording, add missing words, and clarify references.
> > Some examples in no particular order: $[\ldots]$
> - Finally, we went through the entire document to correct typos, improve wording, add missing words, and clarify equations and references as per the examples provided by the reviewers.

---

### Review · Reviewer_n7vN · 2024-01-14

**Summary Of Contributions:**

The paper surveys approaches to the credit assignment problem (CAP) in Deep RL and proposes a unifying framework. Three lines of research in CAP are described: (a) action influence quantification, (b) learning of action influence, and (c) evaluation. Challenges and open problems in the field are laid out together with future research directions.

**Audience:**

Yes

**Claims And Evidence:**

Yes

**Requested Changes:**

See the above section.

**Strengths And Weaknesses:**

Strengths:
* The paper surveys a broad spectrum of material.
* It formulates fundamental questions and proposes a unified view of CAP. This is very useful both for researchers and the people who enter the field of RL.
* The paper provides food for thought, identifies the gaps in the current state-of-the-art, and organizes them in a list of challenges.

Weaknesses:
* The paper has several technical errors that are misleading to various degrees, e.g.:
  * The sum in the definition of $Z_t$ (above eq. (1)) should iterate over $k$.
  * In eq. (8) the items under the curly bracers should be swapped.
  * There seems to be an inconsistency in defining GVF and UFVAs in Section 4.5 and Table 2.
  * Equations (12) and (26) lack the bracer, i.e., $(1 - ...)z$.
  * The intuition at the end of the paragraph "Hindsight advantage" is not clear. Can we have $\pi(a|s)=0$ and $\mathbb P_D(a|s,z)>0$? (it is not clear that this is the case by reading Harutyunyan et al., 2019).
  * Eq. (17) is unclear.
  * It seems that in the sentence directly preceding eq. (30), there should be "for all $V$ and $V'$, \Gamma<1$.
  * It seems that there should be $\hat{V^\pi}$ eq. (30).
* The paper is long with over 43 pages. Some information is duplicated, and the text, while often easy to read, is quite verbose.
* The paper includes the Acknowledgments section, where the initials of the main author together with the grant number are provided. Additionally, a 'thank you' is given to a person mentioned by the first and the last name.
* There are several typos and missing spaces.

---

> ### Author Response · Authors · 2024-02-07
> **Official response to Reviewer n7vN**
>
> Thank you for the thorough review and the detailed comments.
>
> We have carefully gone through the paper and addressed the technical errors listed in the review, together with new others found during the revision. In particular:
> > The sum in the definition of $Z_t$ (above eq. (1)) should iterate over $k$.
> - We have corrected the sum notation above Equation (1), ensuring it iterates over the appropriate set.
>
> > In eq. (8) the items under the curly bracers should be swapped.
> - Adjustments have been made to Equation (8), swapping the items under the curly braces as suggested.
>
> > There seems to be an inconsistency in defining GVF and UFVAs in Section 4.5 and Table 2.
> - We resolved inconsistencies in defining GVF and UFVAs in Section 4.5 and Table 2. Equations now have the same form.
>
> > Equations (12) and (26) lack the bracer, i.e., $(1-\ldots)z$.
> - Corrections were applied to Equations (12) and (26) to include the missing braces.
>
> > The intuition at the end of the paragraph "Hindsight advantage" is not clear. Can we have $\pi(a|s) = 0$ and $\mathbb{P}_D(a|s, z) > 0$? (it is not clear that this is the case by reading Harutyunyan et al., 2019).
> - We have clarified the intuition at the end of the "Hindsight advantage" paragraph. In particular, we use the case you proposed as an example to clarify the description.
>
> > Eq. (17) is unclear
>
> > It seems that in the sentence directly preceding eq. (30), there should be "for all $V$ and ,$V'$, $\Gamma<1$.
>
> > It seems that there should be $\hat{V}^\pi$ eq. (30).
>
> - Clarified the form of Equation (17) and corrected the sentence preceding Equation (30).
>
> Furthermore:
> - We have streamlined the manuscript to eliminate duplication and reduce verbosity of Sections 5 and 6, making the text more concise and focused on key points. The manuscript is now two pages shorter than the previous revision.
> - We removed the acknowledgements section.
> - We proofread the whole document again.

---

### Review · Reviewer_icBr · 2024-01-31

**Summary Of Contributions:**

Summary: This paper writes a sufficient and detailed survey of temporal credit assignment in Deep Reinforcement Learning. The paper firstly formally defines the MDP, the goal, and the credit assignment function. Then the paper discuss the current assignment functions used in most reinforcement learning algorithms. After that the paper introduces several challenges to assign credit in deep RL: delayed effects due to MDP depth, low action influence due to low MDP density, low action influence due to high MDP breadth, and discuss the relationship with the exploration problem. Then the paper classifies RL methods in "Time as a heuristic", "Decomposing return contributions", “Conditioning on a predefined set of goals”, "Modeling transitions as sequences", "Planning and learning backwards", "Meta-learning proxies for credit". Then the paper propose some evaluation metrics to evaluate the credit assignment in deep RL.

**Audience:**

Yes

**Claims And Evidence:**

Yes

**Requested Changes:**

Please clarify about the following points:
1.The paper thinks the action-value function is also a form of credit assignment. But I think it is not true. Learning an immediate reward function that assigns the credit can be called as the credit assignment function. Other forms can not viewed as a valid credit assignment function.
2.The paper does not provide quantitative comparison or analysis of current credit assignment methods.
3.Some definitions that measures the credit assignment function rather than the training performance or value errors.

**Strengths And Weaknesses:**

Comments: 1.The paper clearly defines the concept of credit assignments. 2.The paper thinks the action-value function is also a form of credit assignment. But I think it is not true. Learning an immediate reward function that assigns the credit can be called as the credit assignment function. Other forms can not viewed as a valid credit assignment function. 3.The paper does not provide quantitative comparison or analysis of current credit assignment methods. 4.Some definitions that measures the credit assignment function rather than the training performance or value errors.

---

> ### Author Response · Authors · 2024-02-07
> **Official response to Reviewer icBr**
>
> Thank you for the insightful review. To address the raised concerns:
>
> > The paper thinks the action-value function is also a form of credit assignment. But I think it is not true. Learning an immediate reward function that assigns the credit can be called as the credit assignment function. Other forms can not viewed as a valid credit assignment function.
>
> The survey highlights that, in the current state of the art, there is no shared, formal definition of credit. For this reason, it proposes one of many possible interpretations of it.
> The reason why we argue that state-action values can quantify credit in our framework is that we consider the *return* of a state-action as a valid measure of influence.
> On the other hand, we do note the limitations of such a measure, as it cannot provide a relative form of credit and cannot provide an estimation of the contribution of a single action.
>
> > The paper does not provide quantitative comparison or analysis of current credit assignment methods.
> Indeed. The aim of the paper is to provide a survey of the ways to quantify credit, the methods to do it, and of the protocols to evaluate them. Our claim does not include providing a quantitative comparison between the methods.
> While such a benchmark could be a great addition to the literature, we argue that it is an effort requiring a study of its own and leave this to future work.
>
> Indeed. The aim of the paper is to provide a survey of the ways to quantify credit, the methods to do it, and of the protocols to evaluate them. Our claim does not include providing a quantitative comparison between the methods.
> While such a benchmark could be a great addition to the literature, we argue that it is an effort requiring a study of its own and leave this for future work.
>
> > Some definitions that measures the credit assignment function rather than the training performance or value errors
>
> We kindly ask the reviewer to clarify this concern.

---

### Author Response · Authors · 2024-02-07
**Summary to all reviews**

Dear reviewers,

Thank you for your thorough reviews, the constructive feedback and the comments on the manuscript. We have carefully considered each point raised and have made substantial revisions to address the concerns.

All reviews agreed that the manuscript surveys a broad range of material, that it provides a solid and clear formalisation of the CAP, identifies the gaps in the current state of the art, thoughtfully organises them into challenges and that overall it results in a useful tool in the hands of researchers and new entry practitioners. Most reviews agreed that the paper is well written up to Section 5.

However, most reviews agreed that Sections 6 and 7 do not make the best use of these solid foundations. Each review raised specific issues that we addressed in the response to each reviewer. One review has raised concerns about the length of the document.
Below, we present a brief summary of the main revisions, and also provide an individual response to each review that details how these revisions address the points raised by each reviewer.

In short, we have:
- Better connected Section 6 to the formalisms introduced in Section 4 and 5 with a more rigorous definition of a credit assignment method and more details for each method about the type of assignment used.
- Increased the clarity of Section 5 by removing unnecessary details and more insights in Section 5.4
- Highlighted which measure of influence is used by each class of methods.
- Increased the rigour of the descriptions of the methods in Section 6.
- Refactored the descriptions of the methods to remove redundant information and merged similar methods into a single paragraph.
- Better detailed the evaluation metrics in Section 7
- Fixed the detailed issues raised by reviewers n7vN and 3PaG.

We believe that these revisions have substantially improved the manuscript, addressing the concerns raised by the reviewers while enhancing the clarity, coherence, and contribution of our work.

---

### Comment · Reviewer_3PaG · 2024-02-29
**Final review**

Thank you to the authors for your responses and your changes to the paper. Portions of the paper are improved from the initial submission. There are a lot of useful pieces of information and ideas in this work.

Unfortunately, I am recommending that this work be rejected as it does not meet the level of quality, in terms of correctness and clarity, required of TMLR. I will detail a non-exhausitve list of issues that remain. I believe that the number of small issues and the severity of a small number of issues are too substantial to be resolved in a camera-ready editing phase.

Severe issues:
- I brought up in my initial review that Table 4 contains interesting information but is not referenced in the text and there is no explanation about how the authors arrived at the conclusions in the table. A single reference has been added, but it is merely a pointer and does not contain any discussion or explanation about Table 4's contents. This reference is insufficient and I still do not understand how the table was constructed. In fact, there are no uses of "depth", "density", and "breadth" in Section 6 that refer to the CA challenges in Table 4.

  An additional minor formatting note is that there appear to be missing and misplaced horizontal lines for separating methods of different classes in Table 4.
- I also brought up in my initial review that some of the section summaries were confusing and difficult to read. Section 6.1.4 is still very confusing and difficult to read.
- The equations in Section 6 from Section 6.2 onward are confusing and contain undefined, implicitly ignored, and duplicated symbols. For example, in Equation 19, $K$'s $c$ and $g$ arguments are ignored, it is ambiguous where $s_{t - 1}$, $a_{t - 1}$, $s_t$, and $a_t$ come from, and how they relate to $R^*$'s arguments ($s$ and $a$). For another example, the symbols $c$ and $g$ are each used to refer to two different concepts in Equation 21.

Minor issues:
- In Table 3, advantage should be shown as only partially satisfying the recursive property. The advantage is not naturally recursive, but it can be decomposed into recursive components (the state and action value functions).
- The summaries in Section 6.2.1 and Section 6.3.1 are also still confusing.
- One of the factors that makes this work feel long is that many of the paragraphs are very long and combine the discussions of many different ideas. For example, in Section 4, the state-action values subsection is essentially a single paragraph that takes up half a page. In Section 6.1.1, the AL paragraph could be split in half at the sentence beginning with "Direct Advantage Estimation (DAE)". In Section 6.1.2, the ET paragraph could each be split into three paragraphs and the ETDs paragraph could be split into four. The first paragraph of 6.1.3 could be three paragraphs and the second paragraph in that section could itself be two paragraphs.
- The discussion of exploration in Section 5.4 is much improved but I see two places where the second paragraph could be split into three more focused paragraphs.
- Section 5.5, it states that "we surveyed the literature and collected the issues that arise from solving the CAP". The issues listed are obstacles to solving the CAP, not issues that arise from solving the CAP. In addition, one of the listed items, "resorting to time as a heuristic", is neither type of issue.
- There are places in the text where parenthetical citations are misused as nouns, e.g., "(Baird, 1999) also uses time as a proxy for causality" and "(Oh et al., 2018; Ferret et al., 2021b) uses what can be viewed as an optimistic, empirical estimate of the advantage . . .". In the sentence "Direct Advantage Estimation (DAE) Pan et al. (2022) follows (Wang et al., 2016b) with the same specification . . ." has both missing parentheses around (Pan et al., 2022) and extra parentheses around (Wang et al., 2016b).
- Two expectations in the AL paragraph of Section 6.1.1 have closing parentheses instead of braces.
- Section 6.3, at the end of the GVF subsection, the question "what is the expected return of this action, given that I am going to achieve this goal?" is given as the prediction problem for GVFs. This phrasing is incorrect as conditioning on achieving the goal constrains the type of behavior that can be exhibited after taking action $a$ in state $s$ and assumes that the goal can even be achieved afterward. An alternative phrasing is "what is the expected return of this action given that $g$ is the goal?".

---

### Author Response · Authors · 2024-04-19
**Camera ready revision**

Dear Action Editor,

Thank you for the insightful feedback and the time spent to review our work.
We have carefully addressed all the remaining issues raised by the reviewer 3PaG (comment of the 29th Feb) and have made key revisions to enhance the correctness, clarity, and overall quality of the manuscript.

Specifically:
- Table 4: we substantially expanded the discussion surrounding Table 4, providing a clear explanation of how the conclusions were derived. Each subsection now contains justifications for each choice in the table. We also fixed the formatting issues.
- Confusing Sections: We rewrote the summaries in sections 6.1.4, 6.2.1, and 6.3.1 to improve their clarity and readability.
- Equations: Revised equations from Section 6.2 onward, clarifying definitions of symbols, addressing ambiguity, and correcting inconsistencies.

For the minor issues:
- In Table 3, we ensured the recursive property is accurately represented for advantage;
- We improved, the summaries in Section 6.2.1 and Section 6.3.1, as already explained above;
- We improved the paragraph structure in the whole manuscript, including Section 4, Section 6.1.1, 6.1.2 and 6.1.3.
- We have reviewed, and fixed parenthetical citation errors, parenthesis in equations and grammar throughout the article.

All these edits are in the latest revision uploaded.

We appreciate you recognizing the value of our work in providing a comprehensive survey and unifying framework for the credit assignment problem (CAP) in deep reinforcement learning.
Thank you again for your thorough reviews and for the improvements they brought to the final version of the manuscript.

---

> ### Comment · Action_Editor_e34D · 2024-04-20
> **Missing references**
>
> Dear authors,
>
> I am checking your manuscript, but the references to and inside the Appendix are missing.
> Can you fix the problem?
>
> Best,
> AE

---

> > ### Author Response · Authors · 2024-04-20
> > **Fixed references**
> >
> > Dear AC,
> >
> > Thank you for the prompt response, and apologies for the oversight.
> > We have fixed the references to the Appendix and uploaded the new PDFs for both the main article and the supplementary material.
> >
> > Thank you,
> > Authors

---

### Decision · Action_Editor_e34D · 2024-03-29

**Recommendation:** Accept with minor revision

**Comment:**

This paper reviews the Credit Assignment Problem (CAP) in Reinforcement Learning (RL), focusing on Temporal Credit Assignment (CA) in deep RL. It proposes a unified formalism for credit, addresses challenges such as delayed effects, and suggests evaluation protocols. Overall, it provides insights for newcomers and suggests future research directions in the CAP.

The paper has many strong points: clear definitions, comprehensive survey, formulation of fundamental questions, identification of gaps and challenges, effective setup, interesting discussion, and thoughtful analysis.

On the other hand, the reviewers raised several technical and presentation issues, which were partially addressed by the updated version provided by the authors. After reading the new version of the paper, the reviewers feel that some points were not properly fixed (see the message by reviewer 3PaG posted on February 29th).

The authors need to provide a new version of their paper addressing all the open issues.

**Audience:**

The paper deals with a topic that is of interest to a large part of the TMLR's audience.

**Claims And Evidence:**

This paper is a survey of methods for the credit assignment problem in deep reinforcement learning.
The paper proposes a unifying framework and identifies gaps in the state of the art.
All the contributions are based on an extensive literature analysis.